# BoSS: A Best-of-Strategies Selector as an Oracle for Deep Active Learning

**Denis Huseljic**                                                                          *denis.huseljic@uni-kassel.de*
*Intelligent Embedded Systems*
*University of Kassel*
*Kassel, Hesse, Germany*

**Paul Hahn**                                                                                   *paul.hahn@uni-kassel.de*
*Intelligent Embedded Systems*
*University of Kassel*
*Kassel, Hesse, Germany*

**Marek Herde**                                                                             *marek.herde@uni-kassel.de*
*Intelligent Embedded Systems*
*University of Kassel*
*Kassel, Hesse, Germany*

**Christoph Sandrock**                                                            *christoph.sandrock@tuwien.ac.at*
*Machine Learning Research Unit*
*TU Wien*
*Vienna, Vienna, Austria*

**Bernhard Sick**                                                                         *bernhard.sick@uni-kassel.de*
*Intelligent Embedded Systems*
*University of Kassel*
*Kassel, Hesse, Germany*

**Reviewed on OpenReview:** *https://openreview.net/forum?id=qTs6spvhOS*

## Abstract

Active learning (AL) aims to reduce annotation costs while maximizing model performance by iteratively selecting valuable instances. While foundation models have made it easier to identify these instances, existing selection strategies still lack robustness across different models, annotation budgets, and datasets. To highlight the potential weaknesses of existing AL strategies and provide a reference point for research, we explore oracle strategies, i.e., strategies that approximate the optimal selection by accessing ground-truth information unavailable in practical AL scenarios. Current oracle strategies, however, fail to scale effectively to large datasets and complex deep neural networks. To tackle these limitations, we introduce the Best-of-Strategies Selector (BoSS), a scalable oracle strategy designed for large-scale AL scenarios. BoSS constructs a set of candidate batches through an ensemble of selection strategies and then selects the batch yielding the highest performance gain. As an ensemble of selection strategies, BoSS can be easily extended with new state-of-the-art strategies as they emerge, ensuring it remains a reliable oracle strategy in the future. Our evaluation demonstrates that i) BoSS outperforms existing oracle strategies, ii) state-of-the-art AL strategies still fall noticeably short of oracle performance, especially in large-scale datasets with many classes, and iii) one possible solution to counteract the inconsistent performance of AL strategies might be to employ an ensemble-based approach for the selection.

# 1 Introduction

Despite the era of foundation models (Gupte et al., 2024), most machine learning applications still require carefully annotated domain- or task-specific data (Rauch et al., 2024). Active learning (AL) (Settles, 2009) aims to reduce annotation costs by prioritizing instances that maximize model performance. Selecting which subset of instances to annotate is determined by a selection strategy, which is the most critical element of AL. However, recent studies show that there is no single selection strategy that outperforms every other alternative across different domains, model architectures, and budgets (Munjal et al., 2022; Lüth et al., 2024; Werner et al., 2024). AL aims to maximize model performance, yet most strategies do not directly select instances that optimize this goal. Instead, they rely on performance-related heuristics, which can be suboptimal in some scenarios and highly effective in others. Moreover, once chosen, a selection strategy typically remains fixed throughout the entire AL process, limiting the ability to adapt to distribution shifts caused by iteratively annotating new instances. For example, some strategies, such as TypiClust, work well in early stages of AL, but tend to fail in later stages (Hacohen et al., 2022). This inconsistency underscores the challenge of identifying the instances that yield the greatest performance gains for a given budget.

Nevertheless, it is possible to conceptualize a selection strategy that approximates an optimal selection using a so-called *oracle strategy* that leverages ground truth information, including instance labels or access to the test data. Although this information is unattainable in real AL applications, such an oracle strategy provides a useful diagnostic reference for assessing state-of-the-art selection strategies. Comparing the performance gap between selection strategies and the oracle strategy reveals how far these approaches deviate from the ideal one and whether that deviation is concentrated in early, later, or across all cycles. Moreover, analyzing how the oracle selects data may offer new insights for refining existing strategies or guide the development of new, even more effective ones.

However, approximating the optimal selection strategy is inherently challenging due to the combinatorial explosion in finding the best subset, along with the necessity of model retraining. Although some studies have tried to approximate the optimal strategy (Sandrock et al., 2023; Zhou et al., 2021; Werner et al., 2024), these methods are feasible only for small-scale models and datasets, mainly due to their high computational demands, which arise from assessing the influence of each instance independently instead of in batches. While Sandrock et al. (2023) focuses on kernel-based models with tabular data, Zhou et al. (2021) use only small convolutional and recurrent architectures with small-scale datasets such as Fashion-MNIST (Xiao et al., 2017). Although a more practical strategy was introduced by Werner et al. (2024) recently, it remains computationally expensive for larger budgets. For example, due to high computational costs, the authors extrapolated results for batch sizes above 500. Overall, current oracle strategies do not scale to more challenging, larger datasets, making it impossible to compare them to state-of-the-art AL selection strategies in these settings.

In this article, we propose the *Best-of-Strategies Selector (BoSS)*, a simple and scalable oracle strategy for batch AL that approximates the optimal selection and can be efficiently applied to large-scale deep neural networks (DNNs) and datasets. BoSS first constructs a diverse pool of candidate batches through an ensemble of selection strategies. It then adopts a performance-based perspective, selecting the candidate batch that, once annotated, leads to the highest performance improvement. For efficiency, BoSS freezes the pretrained backbone and assesses the performance improvement of candidate batches by retraining only the final layer during selection. By combining an ensemble-based preselection of candidate batches, a performance-based batch assessment, and a frozen backbone, BoSS serves as a batch oracle strategy that also works in large-scale deep AL settings, something that

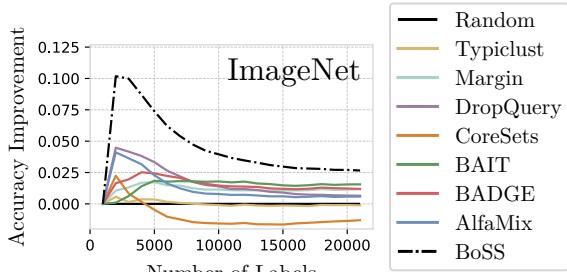

Figure 1: Accuracy improvement over Random sampling for BoSS and state-of-the-art AL strategies using DINOv2-ViT-S/14.

previous oracles do not achieve (cf. Fig. 1). Our experiments on a variety of image datasets demonstrate that i) BoSS outperforms existing oracle strategies under comparable computational constraints, ii) cur-

rent state-of-the-art AL strategies still fall noticeably short of oracle performance, especially in large-scale datasets with many classes, indicating there remains potential for developing stronger strategies, and iii) no single AL strategy consistently dominates across all AL cycles, highlighting the potential for a robust ensemble-driven AL strategy. Our contributions can be summarized as follows:

> **Contributions**
> - **Scalable Oracle:** We introduce BoSS, the first batch oracle strategy scalable to large datasets and complex DNNs. BoSS combines an *ensemble of selection strategies* with a *performance-based selection,* efficiently realized by *retraining only the final linear layer.*
> - **Comprehensive Evaluation:** Extensive experiments demonstrate that i) BoSS outperforms existing oracle strategies and ii) current state-of-the-art AL strategies fall noticeably short of oracle performance. Our implementation is publicly available at `https://github.com/dhuseljic/dal-toolbox`.
> - **Insights into AL Development:** Our analysis highlights i) that observed gaps between selection strategies and oracle performance suggest room for improvement and ii) that ensemble-based AL approaches can be a potential solution for mitigating the inconsistencies of commonly employed single AL strategies.

## 2 Related Work

**Selection strategies** in AL are typically divided into uncertainty- and representativeness-based strategies. While popular uncertainty-based strategies, such as Margin (Settles, 2009) or BADGE (Ash et al., 2020), assume that a selection of difficult (or uncertain) instances improves performance, representativeness-based strategies, such as TypiClust (Hacohen et al., 2022), favor instances that best represent the underlying data distribution. In recent years, a combination of both has proven to work well (Ash et al., 2021; Gupte et al., 2024) because both *heuristics* are partially related to model performance. Furthermore, as it is common to select batches in deep learning, most strategies ensure diversity within a batch through clustering, avoiding the selection of similar instances (Kirsch et al., 2023; Ash et al., 2020; Gupte et al., 2024).

Despite substantial progress in AL, selection strategies can still fail because heuristics such as uncertainty or representativeness do not guarantee performance improvements (Zhao et al., 2021). Consequently, recent studies have increasingly focused on evaluating the **robustness of AL strategies**, revealing significant challenges in identifying the universally best strategy. Munjal et al. (2022) emphasize this difficulty by demonstrating that no single selection strategy consistently outperforms others, with results heavily dependent on experimental conditions and hyperparameter tuning during AL cycles. Similarly, Lüth et al. (2024) and Rauch et al. (2023) further explore these inconsistencies and propose a standardized evaluation protocol, finding that BADGE generally performs best across diverse experimental setups. However, in contrast, Werner et al. (2024) evaluate strategies across multiple domains, including images, text, and tabular data, concluding that Margin yields the most consistent performance improvements. These conflicting findings underscore the lack of coherence in experimental outcomes and emphasize the ongoing challenge of finding a universally best AL strategy.

Given these challenges, it is natural to explore how an optimal selection strategy would look like. For this reason, oracle strategies (or oracle policies) have been introduced. **Oracle strategies** (Zhou et al., 2021; Sandrock et al., 2023) aim to approximate the optimal selection in a feasible time by leveraging ground truth information typically unavailable to conventional AL selection strategies (e.g., access to all labels). Despite their potential, oracle strategies remain underexplored in the literature, and existing methods often struggle with scalability issues. Zhou et al. (2021) introduced an oracle strategy that employs a simulated annealing search (SAS) to identify an optimal selection order given a fixed budget. Even though they achieve impressive performance, the high number of search steps implies high computational cost, limiting its application to large datasets. Sandrock et al. (2023) introduced an iterative, non-myopic oracle strategy that selects instances based on both immediate and long-term performance improvements through a look-ahead approach. In their experiments, they mainly work with tabular data and employ a kernel-based classifier for fast retraining. Recently, Werner et al. (2024) proposed an oracle strategy as part of an AL benchmark, which we refer to as cross-domain oracle (CDO). Their approach greedily selects the instance,

leading to the highest performance gain from a fixed number of randomly chosen instances. If no instance increases test performance, the selection is performed according to Margin. Due to selecting a single instance at a time, this approach requires retraining after each selection, significantly limiting its scalability for larger budgets.

Methodologically, our oracle strategy positions itself between the strategies of Zhou et al. (2021) and Werner et al. (2024). While Zhou et al. (2021) optimize the selection over the entire labeled pool, and Werner et al. (2024) adopt a greedy, single-instance strategy, BoSS focuses explicitly on batch acquisitions. Consequently, our oracle strategy provides a less greedy perspective than (Werner et al., 2024) by considering the collective performance improvement of instances within a batch. Additionally, our oracle searches more efficiently than (Zhou et al., 2021) by focusing on batch-level performance improvements instead of the entire labeled pool (cf. Section 4). Searching for an optimal batch from a large pool is considerably simpler than searching for a (much larger) labeled pool. Consequently, BoSS demonstrates superior scalability, remaining effective even on large-scale datasets such as ImageNet.

## 3 Notation

We consider classification tasks in a pool-based AL setting. Let $\boldsymbol{x} \in \mathcal{X}$ be an instance and $y \in \mathcal{Y} = \{1, \ldots, K\}$ denote its corresponding label, where $K$ is the number of classes. Further, let $\mathcal{U} \subset \mathcal{X}$ be the large unlabeled pool and $\mathcal{L} \subset \mathcal{X} \times \mathcal{Y}$ be the labeled pool. While $\mathcal{U}$ is assumed to be sampled i.i.d. from distribution $p(\boldsymbol{x})$, $\mathcal{L}$ is typically biased towards instances a selection strategy considers informative. Additionally, since our focus is on oracle strategies for evaluation, we consider an evaluation dataset $\mathcal{E}$. At the start of AL, we initialize $\mathcal{L}$ by randomly sampling $b$ instances from $\mathcal{U}$. Then, we perform a total of $A$ AL cycles, selecting $b$ instances to label in each cycle. The total labeling budget is denoted by $B = b + A \cdot b$. As our model, we consider a DNN consisting of a feature extractor $h^{\boldsymbol{\phi}} : \mathcal{X} \to \mathbb{R}^D$ and a classification head $g^{\boldsymbol{\theta}} : \mathbb{R}^D \to \mathbb{R}^K$, where $\boldsymbol{\phi}$ and $\boldsymbol{\theta}$ are trainable parameters. Hence, a DNN is a function $f^{\boldsymbol{\omega}} = (g^{\boldsymbol{\theta}} \circ h^{\boldsymbol{\phi}})(\boldsymbol{x})$ mapping an instance to the logit space, where $\boldsymbol{\omega} = \{\boldsymbol{\phi}, \boldsymbol{\theta}\}$. The conditional distribution $p(y|\boldsymbol{x}, \boldsymbol{\omega}) = [\mathrm{softmax}(f^{\boldsymbol{\omega}}(\boldsymbol{x}))]_y$ is modeled through the output of the DNN. We additionally consider the posterior distribution over parameters $p(\boldsymbol{\omega}|\mathcal{L})$ and the predictive distribution $p(y|\boldsymbol{x}, \mathcal{L}) = \mathbb{E}_{p(\boldsymbol{\omega}|\mathcal{L})}[p(y|\boldsymbol{x}, \boldsymbol{\omega})]$.

## 4 A Formalization of Performance-based Active Learning

The *main goal* of AL is to acquire the labeled pool that minimizes the model's error (or maximizes its performance) on unseen instances. We formalize the corresponding optimization problem by

$$\mathcal{L}^{\star} = \underset{\mathcal{L} \subset \mathcal{U}}{\arg\min} \; \mathbb{E}_{p(\boldsymbol{x}, y)}\big[\ell\big(y, p(y|\boldsymbol{x}, \mathcal{L})\big)\big] \quad \text{subject to} \quad |\mathcal{L}| = B, \tag{1}$$

where $\ell(y, p(y|\boldsymbol{x}, \mathcal{L}))$ denotes a loss function that quantifies the discrepancy between the true label $y$ and the probabilistic prediction $p(y|\boldsymbol{x}, \mathcal{L})$. Note the slight abuse of notation $\mathcal{L} \subset \mathcal{U}$ to signify that instances in $\mathcal{L}$ are seen as a subset of those in $\mathcal{U}$, even though $\mathcal{L}$ includes labels and $\mathcal{U}$ does not. Solving Eq. (1) is computationally infeasible due to the enormous number of possible combinations of instances for $\mathcal{L}$ and, more importantly, the absence of labels. Focusing on oracle strategies, we consider a supervised subset selection problem, i.e., the labels for all instances in $\mathcal{U}$ are accessible to the oracle.

While most AL selection strategies address the optimization problem in Eq. (1) indirectly (e.g., through uncertainty), some traditional strategies aim to optimize this objective directly (Roy & McCallum, 2001). To this end, they employ a greedy approach, simplifying the problem of choosing $\mathcal{L}$ to acquiring a single label per cycle. More specifically, for $B$ cycles, they select the instance $\boldsymbol{x}_c$ for annotation that leads to the lowest error when added to the labeled pool:

$$\boldsymbol{x}^{\star} = \underset{\boldsymbol{x}_c \in \mathcal{U}}{\arg\min} \; \mathbb{E}_{p(\boldsymbol{x}, y)}\big[\ell\big(y, p(y|\boldsymbol{x}, \mathcal{L}^{+})\big)\big], \tag{2}$$

where $\mathcal{L}^{+} = \mathcal{L} \cup \{(\boldsymbol{x}_c, y_c)\}$ is the extended labeled pool. This new optimization problem resolves the combinatorial problem by sequentially extending the labeled pool $\mathcal{L}$.

However, the acquisition of a single instance per cycle poses several challenges when working with DNNs. The greedy selection in Eq. (2) only considers the immediate reduction in error rather than considering the long-term impact of instances (Zhao et al., 2021). This is particularly problematic for DNNs, where retraining with a single additional instance has little influence on the model's predictions (Sener & Savarese, 2018). Furthermore, retraining the model after each label acquisition is computationally impractical, especially in deep learning, where model training can take hours or even days (Huseljic et al., 2025).

To address this problem, we reformulate the optimization problem in Eq. (2) to allow for batch selection. Specifically, over $\lceil B/b \rceil$ cycles, we select a batch $\mathcal{B} = \{\boldsymbol{x}_{c_1}, \ldots, \boldsymbol{x}_{c_b}\}$ of $b$ instances that minimize the error according to

$$\mathcal{B}^\star = \underset{\mathcal{B} \subset \mathcal{U}}{\arg\min} \; \mathbb{E}_{p(\boldsymbol{x},y)}\big[\ell\big(y, p(y|\boldsymbol{x}, \mathcal{L}^+)\big)\big], \tag{3}$$

where $\mathcal{L}^+ = \mathcal{L} \cup \{(\boldsymbol{x}_i, y_i) \mid i = c_1, \ldots, c_b\}$. Although this formulation introduces a combinatorial problem of selecting the batches $\mathcal{B}$, it is simpler than the one in Eq. (1), as batch acquisition sizes are typically much smaller than the labeled pool in deep AL.[1]

While evaluating all possible sets of $\mathcal{B}$ remains infeasible, our idea is to effectively approximate the optimization problem by only considering a *subset of the most promising batches*. The idea of directly selecting a batch from a set of batches is mostly avoided in deep AL (Ash et al., 2021; Hacohen et al., 2022; Gupte et al., 2024). Typically, as heuristic strategies often yield batches with highly similar instances (Kirsch et al., 2023), the batch selection process is simplified through clustering of representations, emphasizing diversity by selecting informative instances from each cluster (Ash et al., 2020; Hacohen et al., 2022; Gupte et al., 2024). This is especially important in early cycles of AL. However, it enforces diversity at every cycle, even when it is not beneficial (Hacohen et al., 2022). In contrast, directly searching for the best batch, as done in Eq. (3), allows the model itself to determine the most effective batch each cycle. While early cycles may benefit from diverse batches, later stages might favor more uncertain and less diverse ones. Additionally, directly considering promising batches rather than instances better captures the instances' long-term impact by evaluating how they collectively influence performance, leading to less greedy behavior.

## 5 An Efficient Oracle Strategy for Deep Neural Networks

We consider the objective in Eq. (3) to build an oracle strategy approximating optimal batch selection that can be efficiently applied in deep learning settings. Our proposed solution can be expressed as follows:

$$\mathcal{B}^\star = \underbrace{\underset{\mathcal{B} \subset \mathcal{U}}{\arg\min}}_{\text{Batch Selection}} \underbrace{\mathbb{E}_{p(\boldsymbol{x},y)}}_{\text{Performance Estimation}} \underbrace{\big[\ell\big(y, p(y|\boldsymbol{x}, \mathcal{L}^+)\big)\big]}_{\text{Retraining}} \equiv \underset{\mathcal{B} \in \{\mathcal{B}_1, \ldots, \mathcal{B}_T\}}{\arg\min} \sum_{(\boldsymbol{x},y) \in \mathcal{E}} \mathbb{1}\big[y \neq \underset{c \in \mathcal{Y}}{\arg\max}\, p(c|\boldsymbol{x}, \mathcal{L}^+)\big] \tag{4}$$

The optimization problem comprises three key components: *Batch selection* involves identifying an optimal batch $\mathcal{B}$ that yields the largest performance improvement, *performance estimation* considers how to evaluate the model's performance when trained with additional data, including evaluation dataset $\mathcal{E}$ and loss function $\ell$, and *retraining* refers to the process of efficiently retraining the DNN and computing updated predictions $p(y|\boldsymbol{x}, \mathcal{L}^+)$. In this section, we focus on how to efficiently implement each of these components.

### 5.1 Batch Selection

As described in Section 4, searching for the optimal batch introduces a combinatorial problem. For example, with an unlabeled pool of 1,000 instances and an acquisition size of 10, the number of possible batches is $\binom{|\mathcal{U}|}{b} = \binom{1000}{10} \approx 2.63 \cdot 10^{23}$, making it computationally infeasible to iterate over all batches. In BoSS, we address this by restricting the search space to a fixed subset of $T \ll \binom{|\mathcal{U}|}{b}$ candidate batches $\{\mathcal{B}_1, \ldots, \mathcal{B}_T\}$.

---

[1]The complexity of this combinatorial problem depends on the acquisition size $b$. By assuming $1 < b \ll \frac{|\mathcal{U}|}{2}$, we obtain a problem that is less complex than the worst-case with $\binom{|\mathcal{U}|}{|\mathcal{U}|/2}$ subsets.

Consequently, the effectiveness of this approximation depends on the particular choice of those candidate batches. A naive approach is to solely draw batches randomly from the unlabeled pool

$$\mathcal{B}_t \sim \text{Unif}\left([\mathcal{U}]^b\right), \tag{5}$$

where $\text{Unif}(\cdot)$ denotes uniform sampling and $[\mathcal{U}]^b$ denotes all possible subsets of $\mathcal{U}$ with size $b$. However, this approach might be inefficient because randomly selecting batches from the unlabeled pool ignores information about the data distribution or the model. In the example above, even if billions of near-optimal batches exist, the probability that a random sample of 100 candidate batches contains one of them is almost negligible.

For this reason, we suggest selecting a set of candidate batches through existing selection strategies. Recent studies (Hacohen et al., 2022; Munjal et al., 2022; Werner et al., 2024) have shown that most strategies lack robustness across varying AL scenarios (e.g., a strategy effective for low budgets may not perform well for higher budgets). Given these insights, we leverage multiple perspectives of a variety of state-of-the-art selection strategies with complementary goals. By incorporating strategies that prioritize diversity or representativeness, we enhance exploration for lower budgets. Similarly, emphasizing uncertainty or model change supports exploitation for higher budgets. Constructing candidate batches $\{\mathcal{B}_1, \ldots, \mathcal{B}_T\}$ in this manner, and then selecting the one that minimizes the error, naturally balances exploration and exploitation. In principle, all strategies from the literature are potentially suitable for our oracle strategy. Furthermore, our oracle is highly flexible, since newly proposed strategies can be seamlessly integrated. Here, we focus on a carefully chosen set of state-of-the-art strategies (cf. Table 1) that are selected based on three jointly considered key criteria:

- **Coverage of relevant heuristics:** The selection strategies encompass all heuristics discussed in Section 2.

- **State-of-the-art performance:** These selection strategies have consistently demonstrated strong performance in research.

- **Efficient computation:** Each strategy is associated with low computational costs to ensure scalability to large-scale datasets with many instances, classes, and/or feature dimensions.

Additionally, for two clustering-based strategies, we also include a supervised version that exploits labels to ensure each cluster corresponds to a class. We found that this is particularly valuable in tasks with suboptimal representations when clustering is difficult.

Preselecting candidate batches helps solving Eq. (3) more effectively, yet the number of batches $T$ that can be considered for the search is constrained by the available *deterministic* strategies. Furthermore, despite their computational efficiency, some selection strategies still can become costly for large unlabeled pools. To address this, we propose to select multiple candidate batches by applying each strategy to randomly sampled candidate pools $\mathcal{C}_1, \ldots, \mathcal{C}_T \subset \mathcal{U}$, each constrained by a maximum pool size $k_{\max}$. When choosing $k_{\max}$, we simply aim to ensure a representative subset of the unlabeled pool $\mathcal{U}$. Beyond that, we found the choice to have little impact on performance and it can mainly be adjusted to improve selection speed. This reduces computational cost and memory requirements while allowing us to increase the number of candidate batches, even for deterministic selection strategies. Furthermore, we found it useful to vary the size of the candidate pools, as some strategies are prone to outliers or selecting similar instances (cf. Appendix B). The proposed algorithm is detailed in Algorithm 1.

## 5.2 Performance Estimation

Evaluating the model performance after retraining with every candidate batch is essential to determine how much the model has improved. In a supervised setting, this evaluation is typically performed using a labeled validation dataset. However, in AL, such labeled validation datasets are typically not available. Consequently, performance estimation becomes an unsupervised problem and requires alternative methods to assess the model's effectiveness. Performance-based selection strategies, such as expected error reduction (Roy & McCallum, 2001), address this challenge by estimating the *expected error* that considers the factorization of the joint distribution $p(\boldsymbol{x}, y)$.

**Algorithm 1** Candidate Batch Selection

**Require:** Batch size $b$, number of batches $T$, selection strategies $\mathcal{S} = \{s_1, \ldots, s_o\}$, maximum candidate pool size $k_{\max}$, unlabeled pool $\mathcal{U}$, labeled pool $\mathcal{L}$, model $\boldsymbol{\omega}$
1:  $\mathcal{B}_{\mathrm{cand}} \leftarrow \emptyset$
2: **for** each selection strategy $s \in \mathcal{S}$ **do**
3:    $\hat{T} \leftarrow \lfloor T/|\mathcal{S}| \rfloor$      ▷ Determine the number of batches per strategy
4:    **for** repeat $\hat{T}$ times **do**
5:       $k \leftarrow \mathrm{Unif}(\{b, \ldots, k_{\max}\})$  ▷ Sample the size $k$ of the candidate pool
6:       $\mathcal{C} \leftarrow \mathrm{Unif}([\mathcal{U}]^k)$      ▷ Sample a candidate pool $\mathcal{C} \subset \mathcal{U}$ of size $k$
7:       $\mathcal{B} \leftarrow s(\mathcal{C}, \mathcal{L}, b, \boldsymbol{\omega})$  ▷ Apply selection strategy $s$ to candidate pool $\mathcal{C}$
8:       $\mathcal{B}_{\mathrm{cand}} \leftarrow \mathcal{B}_{\mathrm{cand}} \cup \{\mathcal{B}\}$      ▷ Extend $\mathcal{B}_{\mathrm{cand}}$ with batch $\mathcal{B}$
9:    **end for**
10: **end for**
11: **return** $\mathcal{B}_{\mathrm{cand}}$

Table 1: Employed selection strategies for sampling candidate batches with their main characteristics. Strategies marked with * use labels as clusters.

| AL Strategy | Unc | Repr | Div |
|---|---|---|---|
| Random (2009) | ✗ | ✓ | ✓ |
| Margin (2009) | ✓ | ✗ | ✗ |
| CoreSets (2018) | ✗ | ✗ | ✓ |
| BADGE (2020) | ✓ | ✗ | ✓ |
| FastBAIT (2024) | ✓ | ✓ | ✓ |
| TypiClust (2022) | ✗ | ✓ | ✓ |
| AlfaMix (2022) | ✓ | ✓ | ✓ |
| DropQuery (2024) | ✓ | ✓ | ✓ |
| TypiClust* (2022) | ✓ | ✓ | ✓ |
| DropQuery* (2024) | ✓ | ✓ | ✓ |

For BoSS, we aim to establish an oracle strategy, i.e., approximating the best possible strategy that leverages all available information. Thus, it is justified to utilize the test split of a given dataset as our evaluation dataset $\mathcal{E}$ to estimate model performance. This ensures that the performance of the retrained model is accurately captured, and that the selected batches indeed result in the highest gain. Additionally, for the loss function $\ell(\cdot)$, the zero-one loss works best. This is because AL strategies are typically evaluated via accuracy learning curves and the zero-one loss directly corresponds to the accuracy. Our experiments in Section 8 show that the Brier score also works well. This is likely due to being a proper scoring rule, leading to a fine-grained assessment of probabilistic predictions (Ovadia et al., 2019).

### 5.3 Retraining

Retraining, particularly with DNNs, is the most time-consuming step in performance-based AL. Generally, batch selection is employed to avoid frequent training after each AL cycle. In our oracle strategy, however, the DNN is to be retrained for each candidate batch, resulting in $T$ retraining repetitions per selection. Although this is faster than retraining after a single instance, the computational overhead is still considerable and limits the size and the number of candidate batches that can be evaluated. For larger-scale datasets such as ImageNet, this process gets increasingly expensive as the labeled pool $\mathcal{L}$ grows, making naive retraining with each candidate batch computationally infeasible. Moreover, retraining must accurately reflect changes in $\mathcal{L}$ to capture which batches truly improve performance. Specifically, even small changes in $\mathcal{L}$ can considerably alter the training dynamics of large DNNs (e.g., change of optimal hyperparameters) potentially yielding noisy and unreliable performance estimates.

For this reason, we propose a selection-via-proxy approach (Coleman et al., 2020) that decouples the retraining process during the selection from the usual cyclic training in AL. Specifically, we freeze the feature extractor's parameters $\boldsymbol{\phi}$ and only retrain the final linear layer $\boldsymbol{\theta}$. This not only significantly reduces retraining time but also enhances stability by restricting parameter updates to a much simpler model. Furthermore, to assess the candidate batches during the selection, we reduce the number of retraining epochs from 200 (as used in our experiments *after* selection) to 50. As shown in Section 8, this is sufficient to identify influential candidate batches while reducing computation substantially.

While this approach enables the efficient use of BoSS, there are additional approaches to improve retraining efficiency. For instance, by employing continual learning strategies (Huseljic et al., 2025), the retraining time of the DNN scales only with the new batch $\mathcal{B}$, making the process largely independent of the size of the extended dataset $\mathcal{L}^+$. As these approaches involve new training hyperparameters, we opt for the simpler variant of retraining only the last layer and leave the exploration of more complex alternatives for future work.

Table 2: Summary of time complexities of oracle strategies.

| Oracle Strategy | Time Complexity per Batch | Recommended Hyperparameters | # Processed Training Instances |
|---|---|---|---|
| CDO (Werner et al., 2024) | $\mathcal{O}(m \cdot \sum_{i=1}^{b} \texttt{train-eval}(\boldsymbol{\theta}, \mathcal{L}^{+i}, \mathcal{E}))$ | $m = 20$ | $20 \cdot (b \cdot |\mathcal{L}| + \frac{b(b+1)}{2})$ |
| SAS (Zhou et al., 2021) | $\mathcal{O}((s + g) \cdot A \cdot \texttt{train-eval}(\boldsymbol{\theta}, \mathcal{L}^{+b}, \mathcal{E}))$ | $s = 1{,}250, g = 250$ | $1{,}500 \cdot A \cdot (|\mathcal{L}| + b)$ |
| BoSS | $\mathcal{O}(T \cdot \texttt{train-eval}(\boldsymbol{\theta}, \mathcal{L}^{+b}, \mathcal{E}))$ | $T = 100$ | $10 \cdot |\mathcal{S}| \cdot (|\mathcal{L}| + b)$ |

## 6  Comparison of Time Complexity

We investigate BoSS's time complexity of selecting a batch in comparison to existing oracle strategies. Specifically, we consider SAS (Zhou et al., 2021) and the recently introduced CDO (Werner et al., 2024). The time complexities in $\mathcal{O}$-notation are summarized in Table 2, where $\texttt{train-eval}(\boldsymbol{\theta}, \mathcal{L}, \mathcal{E})$ denotes the cost of retraining model $\boldsymbol{\theta}$ on dataset $\mathcal{L}$ and then evaluating it on dataset $\mathcal{E}$. Since all oracle strategies primarily differ in terms of retraining and evaluation frequency, we also report the hyperparameters recommended by the respective approaches along with the total number of training instances processed during selection.

CDO (Werner et al., 2024) greedily selects the instance with the highest performance improvement from $m$ randomly sampled instances. This requires $b \cdot m$ retrainings for a batch of size $b$. Due to its greedy nature, CDO acquires instances sequentially and retrains each time on a labeled pool expanded by one instance, denoted as $\mathcal{L}^{+i}$. SAS (Zhou et al., 2021) performs simulated annealing and greedy refinement search steps, represented by parameters $s$ and $g$, respectively. Their approach requires $s + g$ retrainings, where the labeled pool $\mathcal{L}^{+b}$ has been extended by a batch of $b$ instances. Additionally, as SAS evaluates the entire learning curve at each search step (rather than the improvement of a batch), retraining and evaluation times are multiplied by the total number of AL cycles $A$. In contrast to these strategies, BoSS depends solely on the number of candidate batches $T$, determined by the number of batches per strategy $s \in \mathcal{S}$. Consequently, the retraining frequency remains independent of batch size $b$ and the number of cycles $A$, offering a significant advantage in terms of scalability.

For CDO, Werner et al. (2024) recommend setting $m = 20$, resulting in 20 retrainings per instance selection within a batch. This quickly becomes infeasible with larger batch sizes that are common for more complex datasets requiring higher budgets. For example, a batch size of $b = 100$ would require to retrain 2,000 times, which becomes especially expensive towards the end of the AL process, as the labeled pool $\mathcal{L}$ increases in size. Similarly, SAS recommends $s = 25{,}000$ simulated annealing steps and $g = 5{,}000$ greedy refinement steps. However, these parameters determine the frequency of retrainings to obtain the final optimized pool, i.e., $|\mathcal{L}| = B$. As we compare the frequency of retraining per batch, we divide these values by the total number of AL cycles used in our experiments ($A = 20$). This results in $20 \cdot 1{,}500$ retrainings, which is even less scalable to larger datasets. Finally, considering the number of processed training instances, CDO scales quadratically with batch size, posing a major bottleneck. For instance, with $b = 50$ and $|\mathcal{L}| = 50$, CDO processes approximately 75k instances. In contrast, SAS and BoSS scale linearly, with BoSS achieving a substantially more efficient search, requiring only 11k instances in the same setting.

While our approach involves less frequent retraining, it additionally requires preselecting candidate batches using specific selection strategies. This step introduces extra computational overhead for batch selection. However, with the set of efficient selection strategies we proposed in Table 1, combined with sampling candidate pools significantly smaller than the entire unlabeled pool, this computational burden remains negligible (cf. Section 7). Nevertheless, it is important to highlight that when extending BoSS with more contemporary selection strategies, one must ensure that the computational cost associated with these strategies is considered. In our experiments, we analyzed how the number of candidate batches per strategy affects performance and, in general, increasing this number will always lead to an improvement. However, in Section 8, we found that increasing the number of candidate batches beyond 10 did not yield notable benefits. Thus, we adopt 10 batches per strategy, resulting in a total of $T = 100$ candidate batches.

# 7 Empirical Evaluation of BoSS: Oracle-Level and State-of-the-art AL Comparisons

We evaluate our oracle strategy for the task of image classification. After detailing the experimental setup, we begin with a comparison of BoSS to other oracle strategies. Afterward, we benchmark our approach against state-of-the-art selection strategies across ten image datasets. Our evaluation is driven by four research questions:

$RQ_1$: *Given comparable computational resources, can BoSS match or exceed the accuracy improvements of state-of-the-art deep AL oracle strategies (CDO, SAS)?*

$RQ_2$: *Does BoSS consistently match or surpass the highest test accuracy that any current state-of-the-art AL strategy achieves at every cycle, making it a practical oracle strategy?*

$RQ_3$: *Where lies the greatest potential for improving state-of-the-art AL strategies when comparing them to BoSS across cycles, datasets of varying complexity, and different models?*

$RQ_4$: *What insights regarding AL research can we get by analyzing which selection strategy's candidate batch has been chosen by BoSS?*

In a nutshell, we find that BoSS not only ties or outperforms CDO/SAS in most settings ($RQ_1$) but also serves as a reliable oracle strategy ($RQ_2$), with the biggest performance improvements appearing in large-scale multiclass settings ($RQ_3$). Moreover, our results highlight that each AL strategy contributes to the selection of BoSS and that there is no single best strategy across datasets or cycles within a dataset ($RQ_4$), emphasizing potential advantages in using an ensemble-based AL approach that combines multiple strategies (Donmez et al., 2007).

## 7.1 Experimental Setup

We evaluate our oracle strategy on ten image datasets of varying complexity. For each dataset, we conduct 20 AL cycles, starting with a randomly selected initial pool of $b$ instances, and selecting an additional batch of $b$ new instances in each subsequent cycle. Batch sizes were determined by analyzing the convergence of learning curves obtained via Random sampling. Consequently, the complexity of each dataset is indicated not only by the number of classes $K$ but also by the respective batch size. Table 3 summarizes these datasets, their number of classes $K$, and the employed AL batch sizes $b$.

We employ two pretrained Vision Transformers (ViTs) (Dosovitskiy et al., 2020) that are complemented by a randomly initialized fully connected layer. Specifically, we use DINOv2-ViT-S/14 (Oquab et al., 2024) (22M parameters) and SwinV2-B (Liu et al., 2022) (88M parameters), whose final hidden layers provide feature dimensions of $D = 384$ and $D = 1024$, respectively.

The two differ both in size and in training paradigm: The former was trained via self-supervised learning, while the latter was pretrained on ImageNet in a supervised manner. Note that the ImageNet results obtained with the SwinV2-B backbone are not fully representative, as the model was pretrained on the same dataset. Nevertheless, we include them for completeness. After a batch is selected, each DNN is trained by fine-tuning the last layer on frozen representations for 200 epochs, employing SGD with a training batch size of 64, a learning rate of 0.01, and weight decay of 0.0001. In addition, we utilize a cosine annealing learning rate scheduler. These hyperparameters were determined empirically across datasets by investigating the loss convergence on validation splits. Note that the number of epochs here applies to training after an AL cycle once a batch is selected. In contrast, the *retraining* epochs described in Section 4 refer to those we use to assess candidate batches.

Table 3: Overview of datasets, showing number of classes $K$ and batch size $b$.

| Dataset | # Classes ($K$) | Batch Size ($b$) |
|---|---|---|
| CIFAR-10 (2009) | 10 | 10 |
| STL-10 (2011) | 10 | 10 |
| Snacks (2021) | 20 | 20 |
| Flowers102 (2008) | 102 | 25 |
| Dopanim (2024) | 15 | 50 |
| DTD (2014) | 47 | 50 |
| CIFAR-100 (2009) | 100 | 100 |
| Food101 (2014) | 101 | 100 |
| Tiny ImageNet (2015) | 200 | 200 |
| ImageNet (2015) | 1000 | 1000 |

To evaluate the AL process, we examine the resulting learning curves of oracle and selection strategies. These include *relative learning curves*, which represent the accuracy difference of each strategy compared

to Random sampling, and the *area under the absolute learning curves* (AULC). The corresponding absolute learning curves can be found in Appendix F. All reported scores are averaged over ten trials. For visual clarity, standard errors have been omitted from the figures. All benchmark experiments were conducted on servers equipped with NVIDIA Tesla V100 and A100 GPUs as well as AMD EPYC 7742 CPUs. Experiments involving runtime measurements were performed on a workstation with an NVIDIA RTX 4090 GPU and an AMD Ryzen 9 7950X CPU in a controlled environment to ensure reproducibility and minimize external influences on the measurements (e.g., by ensuring consistent CPU load and hardware configuration).

## 7.2 Benchmark Results

To answer $RQ_1$, we first align hyperparameters of CDO and SAS to closely match the empirical runtime of BoSS, and then compare the resulting learning curves. In principle, with a longer runtime we consider more combinations to solve the combinatorial problem, inevitably improving each oracle's performance. Accordingly, we ensure a fair comparison by approximately equalizing runtimes. Due to the high computational effort of oracle strategies, we focus on four datasets using the DINOv2-ViT-S/14 model. When aligning hyperparameters, we made sure that CDO and SAS have at least as much compute as BoSS, ensuring that any performance advantage is not due to differing computational resources. Importantly, all oracle strategies use the same retraining procedure, ensuring differences in performance are solely due to the selection mechanism. The employed hyperparameter settings are summarized in Table 4, while the associated empirical runtimes can be found in Table 5. Note that different configurations of $\mathcal{S}$ can yield varying runtimes and outcomes. Rather than selecting the ensemble of strategies to optimize runtime, we include all strategies from Table 1 to prioritize robustness across settings (e.g., datasets and models).

Table 4: Hyperparameters of oracle strategies under runtime constraints equivalent to BoSS.

| Oracle | Default | CIFAR-10 ($b = 10$) | Snacks ($b = 20$) | Dopanim ($b = 50$) | DTD ($b = 50$) |
|---|---|---|---|---|---|
| BoSS | $T = 100$ | $T = 100$ | $T = 100$ | $T = 100$ | $T = 100$ |
| CDO | $m = 20$ | $m = 20$ | $m = 10$ | $m = 4$ | $m = 3$ |
| SAS | $s = 25{,}000, g = 5{,}000$ | $s = 250, g = 10$ | $s = 225, g = 10$ | $s = 215, g = 10$ | $s = 150, g = 10$ |

Figure 2 presents the learning curves reporting the relative accuracy improvement of each oracle strategy over Random sampling. Absolute learning curves together with learning curves that report the performance using default hyperparameters can be found in Appendix F. All oracle strategies outperform random selection in terms of accuracy. However, unlike the other oracles, SAS yields only marginal accuracy improvements as the number of search steps was reduced considerably in comparison to the authors' recommendation. Restoring the recommended number improves accuracy but at the cost of much higher computation. In contrast, CDO and BoSS are much more effective, achieving approximately 20% accuracy improvement on CIFAR-10 and Snacks, and around 10% on Dopanim and DTD. We see that especially in larger scale settings with higher batch sizes, BoSS outperforms the other oracle strategies. As shown by the hyperparameters of CDO in Table 4, increasing the AL batch size ($b = 50$) results in a considerable reduction of its hyperparameter ($m = 4$ and $m = 3$). Since $m$ denotes the number of randomly sampled instances from which CDO selects the best, further increasing the batch size would prevent aligning

Table 5: Empirical runtimes of oracle strategies with adapted and default hyperparameters.

| Oracle | Cifar-10 | Snacks | Dopanim | DTD |
|---|---|---|---|---|
| BoSS | 10:07 | 13:19 | 30:47 | 22:07 |
| CDO (Adapted) | 10:26 | 14:11 | 33:25 | 24:56 |
| SAS (Adapted) | 10:20 | 13:24 | 31:56 | 22:14 |
| CDO (Default) | 10:26 | 28:22 | 2:47:08 | 2:46:14 |
| SAS (Default) | 17:22:39 | 24:19:42 | 61:52:42 | 62:34:13 |

its runtime to that of BoSS, highlighting CDO's inefficiency with larger batch sizes. *Overall, BoSS consistently matches or surpasses the performance of other competing oracle strategies across all datasets under comparable computational resources.*

To answer $RQ_2$, we consider the learning curves in Figure 3, depicting relative accuracy improvements over Random sampling. They clearly demonstrate that BoSS consistently outperforms existing AL strategies across all cycles and datasets, for both DINOv2 and SwinV2. *Consequently, we assume BoSS to be a reliable oracle strategy for deep AL selection strategies.* Notably, while the overall accuracy improvement provided

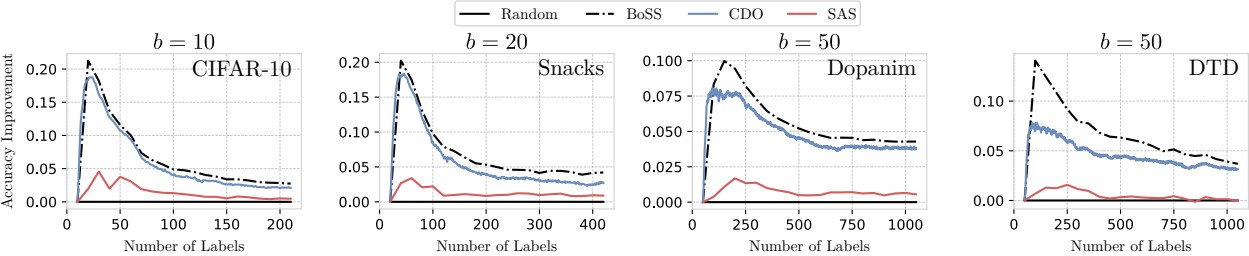

Figure 2: Relative learning curves of oracle strategies with aligned runtimes using DINOv2-ViT-S/14.

by BoSS across the entire AL cycle is modest for simpler datasets such as CIFAR-10 and STL-10, it becomes substantially more pronounced with more challenging datasets, particularly those with more than 20 classes. To complement these findings, we further provide a detailed budget-regime analysis in Appendix G and additional experiments on text data in Appendix H.

To answer RQ$_3$, we compare BoSS to the best-performing AL strategies per dataset. In Fig. 3, we see substantial accuracy differences across all stages of AL. Taking ImageNet with DINOv2 as an example, BoSS achieves approximately twice the accuracy improvement compared to the best-performing AL strategy. The significant gap during the initial exploration phase suggests that there is potentially still room to address the cold-start problem more effectively. Similarly, the performance gap at later cycles indicates

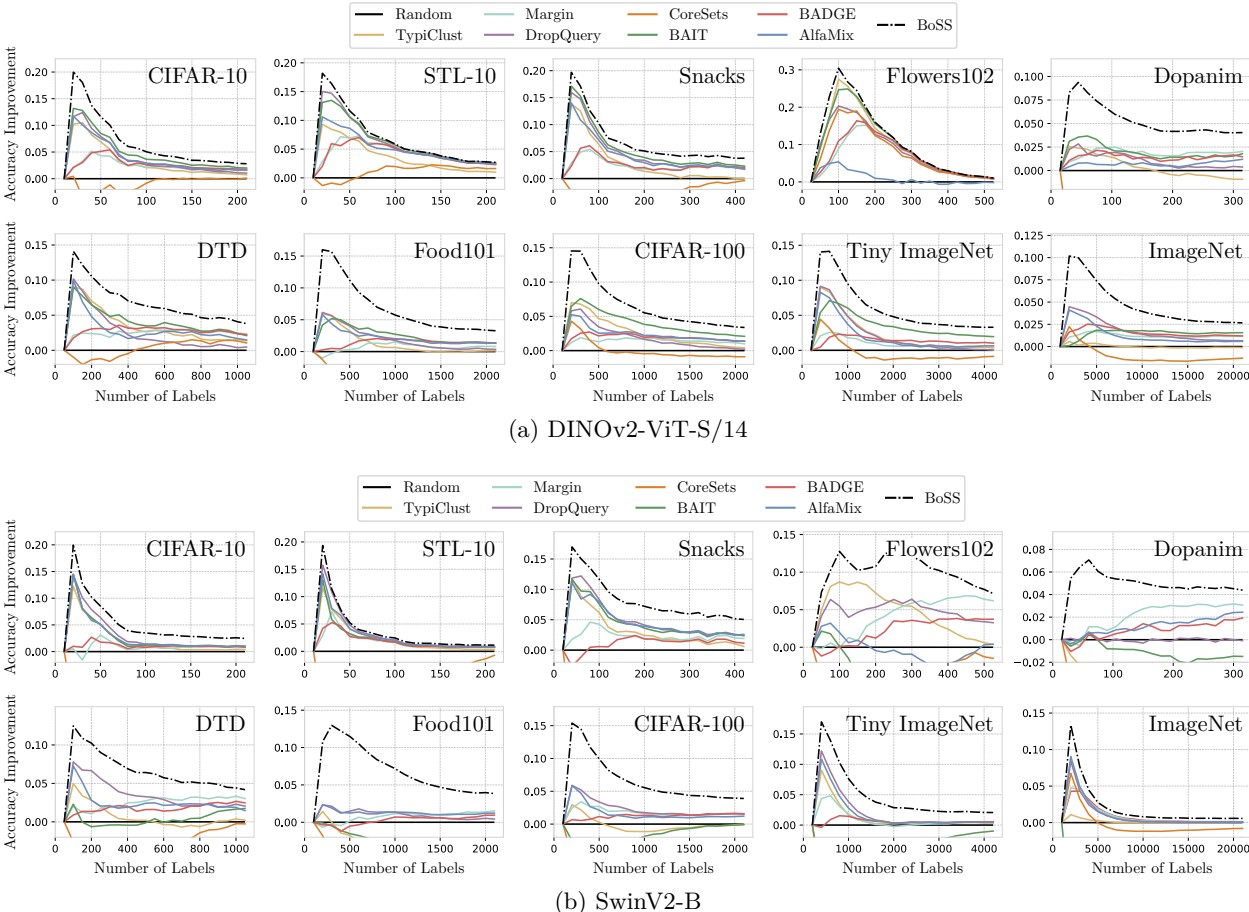

Figure 3: Relative learning curves achieved by BoSS and state-of-the-art selection strategies at each annotation cycle for different pretrained models.

potential shortcomings during the exploitation phase, suggesting that current state-of-the-art AL strategies may struggle to effectively identify instances most valuable for further refinement. Comparisons across datasets reveal that the potential for improving AL strategies correlates with dataset complexity. For example, on less complex datasets such as CIFAR-10, STL-10, or Snacks, AL strategies generally perform closer to the oracle. Conversely, more challenging datasets such as Food101, CIFAR-100, Tiny ImageNet, and ImageNet exhibit substantial gaps. *This finding indicates that large-scale multiclass settings may be a particularly relevant area for further study of AL strategies.* Finally, examining AL strategies across different models demonstrates variability in their effectiveness. For instance, strategies that perform closely to the oracle with DINOv2 on datasets like Snacks or Flowers102 exhibit notably larger performance gaps when used with SwinV2. *This discrepancy suggests that further work on more robust and model-agnostic AL strategies could be beneficial for achieving consistent performance across diverse models.* Alternatively, any newly proposed strategy should include detailed analyses of its failure cases (e.g., being limited to a specific architecture), enabling practitioners to understand the specific scenarios in which it may underperform.

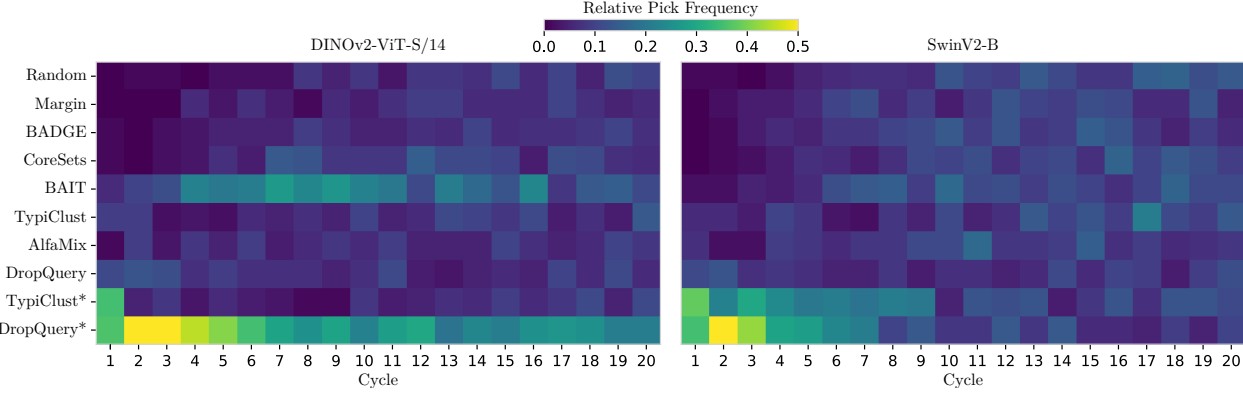

Figure 4: Average relative pick frequency of AL strategies by BoSS across cycles averaged over all datasets.

Finally, to answer RQ$_4$, we examine which AL strategy was selected by BoSS across cycles. For this purpose, Fig. 4 depicts the relative pick frequency of the strategies for both models, each averaged over all datasets. We notice that each strategy is considered at a certain point in the AL cycle, which indicates their respective contribution to the overall performance of BoSS. However, especially at the beginning of AL, the strategies DropQuery* and TypiClust* with supervised cluster assignments dominate, underscoring the importance of representative instances at that stage. Considering the more detailed dataset-specific pick frequencies in Appendix D reveals that this effect is particularly pronounced for large-scale multiclass settings. This suggests that more sophisticated AL strategies may be required in these scenarios and that the supervised selection strategies play an important role for the performance of BoSS. Nevertheless, other strategies such as BAIT are also selected regularly. Notably, towards the end of the AL process, as we approach the convergence region of the learning curves, Random sampling is increasingly taken into account. This suggests that in certain stages none of the specialized AL strategies provide effective candidate batches, which is why more investigation of strategies for exploitation could be beneficial. Similarly, the tendency towards the supervised selection strategies such as DropQuery in the beginning of AL indicates that current strategies struggle to identify effective batches for exploration. *However, most importantly, no single AL strategy consistently outperforms others across all phases of AL. This indicates that the best strategy at a given cycle can vary significantly depending on the context and stage of the AL process.* Consequently, we believe that advancing AL research requires a stronger focus on ensemble-based AL strategies (Donmez et al., 2007; Hacohen & Weinshall, 2023). These strategies integrate multiple strategies and adaptively select the most suitable strategy for the current context, thereby leveraging the specific strengths of each individual strategy.

## 8 Analytical Evaluation: Ablations and Sensitivity Analyses

To better understand the contributions of each component in Eq. (4), we conduct a series of ablation studies and experiments on a representative subset of datasets while fixing the backbone to DINOv2-ViT-S/14. These

analyses aim to isolate the impact of each design choice, evaluate robustness under different conditions, and provide deeper insight into the mechanisms that determine performance.

## 8.1 Selection of Candidate Batches

We compare the proposed selection of candidate batches from Algorithm 1 against the naive selection from Eq. (5). Unlike Random sampling, which directly selects $\mathcal{B}^\star$ at random, the naive selection generates the candidate batches randomly. Relative learning curves for CIFAR-10 and DTD are shown in Figure 5. While the naive selection of candidate batches leads to a better performance than Random sampling, Algorithm 1 considerably improves performance, indicating the importance of a proper candidate batch selection. Moreover, as shown in Table 6, increasing the number of candidate batches $T$ yields further performance gains. Since improvements beyond 10 batches per strategy were negligible, we opted for this value in our experiments, resulting in a total of $T = 100$ candidate batches. Finally, varying the size of candidate pools leads to further improvements, as demonstrated in Appendix B.

Additionally, we examine the influence of selection strategies $\mathcal{S}$. For this, we first applied BoSS with all selection strategies from Table 1 on CIFAR-10, analyzing the frequency with which each strategy's candidate batch was selected. This analysis enables us to identify the most influential strategies specifically for CIFAR-10. Subsequently, we iteratively applied BoSS to the Dopanim dataset, progressively incorporating the next most influential strategy according to the order established earlier. This way, we avoid data-specific overfitting since, in reality, the optimal order for a given task is unknown before running the experiments. Throughout this process, we maintain

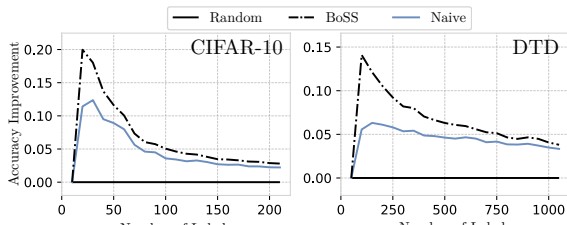

Figure 5: Relative learning curves of BoSS with naively selected candidate batches vs. our algorithm.

a constant total candidate batch size $T$. Consequently, the addition of each new strategy proportionally decreased the number of candidate batches generated by previously included strategies. As shown in Table 8, this sequential inclusion of strategies consistently improves the resulting AULC. These findings suggest that integrating more strategies generally enhances overall performance, and thus, we opt to include a variety of strategies when using BoSS. Intuitively, the more selection strategies we incorporate, the better the robustness of BoSS should be across models, datasets, and domains. Importantly, including additional strategies does not degrade performance, suggesting that BoSS is robust to weaker strategies in the ensemble.

Finally, we examine the impact of batch size $b$, which directly affects the search space of potential candidate batches. As $b$ increases, the number of possible subsets generated from $\mathcal{U}$ grows, making the combinatorial problem in Eq. (3) more difficult. To investigate this impact, we run BoSS with different batch sizes $b$ on the same datasets multiple times. The AULCs in Table 7 on CIFAR-10 demonstrate that increasing the batch size by a factor of four ($4b = 40$) results in a marginal performance decrease. Similarly, for DTD, this batch size ($4b = 200$) yields a slightly more noticeable decline from 67.95 to 66.82. These findings highlight that the effectiveness of solving the combinatorial problem and identifying optimal candidate batches remains sensitive to the chosen batch size $b$. Nonetheless, the results presented in Section 7 show a substantial improvement over all considered state-of-the-art strategies. To further enhance BoSS with even larger batch sizes, increasing the number of candidate batches $T$ seems to be an effective strategy to mitigate potential performance losses.

## 8.2 Estimation of Performance

Here, we examine how different loss functions $\ell(\cdot)$ affect the performance of BoSS. In principle, the choice of $\ell$ will influence the estimation of the performance gain of potential candidate batches. Next to the zero–one loss, which corresponds to the accuracy and therefore has direct relevance for our target metric, we also evaluate two proper scoring rules: cross-entropy and Brier score (Zhao et al., 2021). Similar to the zero–one loss, both loss functions correlate with accuracy, but also quantify the model's probabilistic calibration. As

Table 6: AULC of BoSS with varying number of batches per strategy.

| Batches per Strategy | CIFAR-10 | DTD |
|---|---|---|
| 1 | $89.90_{\pm 0.11}$ | $70.45_{\pm 0.12}$ |
| 5 | $90.45_{\pm 0.10}$ | $71.55_{\pm 0.14}$ |
| 10 | $90.70_{\pm 0.11}$ | $71.79_{\pm 0.11}$ |
| 20 | $90.83_{\pm 0.15}$ | $71.91_{\pm 0.15}$ |

Table 7: AULC of BoSS for different batch sizes.

| Batch Size | CIFAR-10 | DTD |
|---|---|---|
| $0.5 \cdot b$ | $85.71_{\pm 0.32}$ | $68.41_{\pm 0.14}$ |
| $b$ | $85.62_{\pm 0.30}$ | $67.95_{\pm 0.16}$ |
| $2 \cdot b$ | $85.36_{\pm 0.31}$ | $67.37_{\pm 0.16}$ |
| $4 \cdot b$ | $84.95_{\pm 0.30}$ | $66.82_{\pm 0.12}$ |

Table 8: AULC on Dopanim when incorporating additional AL strategies.

| Strategies | AULC |
|---|---|
| Random | $75.24_{\pm 0.15}$ |
| +DropQuery | $75.82_{\pm 0.17}$ |
| +AlfaMix | $76.01_{\pm 0.16}$ |
| +TypiClust | $76.08_{\pm 0.17}$ |
| +BAIT | $76.28_{\pm 0.16}$ |
| +CoreSet | $76.29_{\pm 0.20}$ |
| +Margin | $76.26_{\pm 0.18}$ |
| +BADGE | $76.35_{\pm 0.18}$ |
| +DropQuery* | $76.48_{\pm 0.20}$ |
| +TypiClust* | $76.52_{\pm 0.18}$ |

Table 9: AULC of BoSS using different loss functions.

| Loss ($\ell$) | CIFAR-10 | DTD |
|---|---|---|
| Zero-one Loss | $90.70_{\pm 0.11}$ | $71.79_{\pm 0.10}$ |
| Cross Entropy | $90.53_{\pm 0.12}$ | $71.21_{\pm 0.13}$ |
| Brier Score | $90.67_{\pm 0.10}$ | $71.79_{\pm 0.18}$ |

Table 10: AULC of BoSS across varying numbers of retraining epochs.

| # Epochs | CIFAR-10 | DTD |
|---|---|---|
| 5 | $90.00_{\pm 0.10}$ | $70.77_{\pm 0.11}$ |
| 10 | $90.57_{\pm 0.12}$ | $71.13_{\pm 0.13}$ |
| 25 | $90.71_{\pm 0.13}$ | $71.62_{\pm 0.12}$ |
| 50 | $90.70_{\pm 0.11}$ | $71.80_{\pm 0.11}$ |
| 100 | $90.67_{\pm 0.12}$ | $71.72_{\pm 0.11}$ |
| 200 (Full) | $90.60_{\pm 0.12}$ | $71.84_{\pm 0.07}$ |

a result, they not only give insights about the performance, but also measure the reliability of the predicted probabilities. Table 9 shows that zero-one loss, Brier score and cross-entropy yield similar performance, with cross-entropy slightly lagging behind. For BoSS, we opted for the zero-one loss due to its link to the accuracy, but we can equally use proper scoring rules such as the Brier score. Thus, BoSS is also suitable for scenarios where probabilistic calibration might be important. Accordingly, when employing BoSS, we recommend selecting the metric of interest appropriate for the task at hand.

### 8.3 Retraining

To lower retraining cost within BoSS, we adopt a selection-via-proxy approach, which involves assessing candidate batches by exclusively retraining the final layer for 50 epochs. The impact of different numbers of retraining epochs on the performance of BoSS is detailed in Table 10. We see that reducing this number yields nearly identical AL performance compared to utilizing full retraining with 200 epochs. Consequently, these results suggest that a reduced number of retraining epochs during the selection is sufficient to identify effective candidate batches. Since we recognize a slight decrease in performance going from 10 to 5, and we want to ensure reliable candidate batch assessment, we select 50 retraining epochs as the default for BoSS. However, we additionally investigate the scenario of constructing a highly efficient oracle. To this end, in Appendix E, we show how BoSS performs when both the number of candidate batches and the number of retraining epochs are greatly reduced.

## 9 Limitations

A central limitation of all oracle strategies, including BoSS, is that they should not be interpreted as upper baselines for selection strategies. In particular, the gap observed between BoSS and actual AL strategies cannot be viewed as a directly attainable performance margin. This is because oracles rely on supervised information during selection, making them infeasible in practice and fundamentally distinct from achievable benchmarks. Consequently, the observed performance gap can be conceptually decomposed into two parts: (i) weaknesses of current AL strategies that could, in principle, be improved, and (ii) advantages that stem purely from supervised knowledge and are therefore unattainable. A principled decomposition of these two components remains an open research challenge. Nevertheless, the insights from our evaluation remain valuable, as a larger observed gap still plausibly indicates potential for improvement, even if the true attainable portion of this gap cannot yet be precisely quantified.

In this regard, we further emphasize that the strong performance of BoSS is not solely due to oracle knowledge, since batches are proposed by existing unsupervised AL strategies. Consequently, whenever BoSS identifies a high-performing batch, at least one underlying strategy must already have been capable of proposing it. This property suggests that, in principle, an improved selection may be conceptualized by

learning to identify such high-quality batches. To investigate the importance of supervision in BoSS, we conducted additional experiments (Appendix I) evaluating its performance without access to supervised strategies or ground-truth labels.

## 10   Conclusion

We introduced BoSS, an efficient oracle strategy for batch AL that scales with large datasets and complex DNNs. BoSS achieves a tractable approximation of the optimal selection by: (i) restricting the search space to candidate batches through an ensemble of selection strategies, (ii) assessing performance improvements of those batches by retraining only the final layer, and (iii) selecting batches with the highest performance improvement. Our experiments on ten image classification datasets demonstrate that BoSS outperforms existing oracle strategies and consistently exceeds the performance of state-of-the-art AL selection strategies. Notably, the largest performance improvements emerge on large-scale multiclass datasets, suggesting that these settings are a promising direction for research and for exploring robust, model-agnostic batch selection strategies. The analysis of which selection strategies were chosen showed that BoSS uses a wide range of selection strategies over several AL cycles to achieve both high performance and robustness. This suggests that future AL strategies may increasingly focus on ensemble-based approaches (Huseljic et al., 2026), which, ideally, automatically identify and apply the most appropriate selection strategy in a given cycle.

Although we focus on DNNs in this work, BoSS can easily be combined with other machine learning models. For example, kernel-based approaches are particularly well suited, as retraining can be performed easily and efficiently by updating the kernel matrix. Furthermore, since BoSS consists of an ensemble of selection strategies, it can be easily extended to include new, state-of-the-art AL strategies. As a result, it will continue to provide a reliable oracle strategy in future research. In this context, we envision BoSS as a practical proxy for assessing where current strategies may still have room for improvement. Specifically, whenever a new selection strategy is introduced, we recommend integrating it directly into the ensemble of BoSS. At the same time, we suggest to include the authors' existing (already implemented) comparison strategies as well. This setup provides a straightforward, efficient way to establish an oracle strategy against which novel strategies can be systematically evaluated. An exemplary study demonstrating this procedure can be found in Appendix A.

For future work, a promising direction to improve BoSS is the incorporation of a self-adaptive component that dynamically emphasizes strategies producing high-quality batches. Specifically, this could be achieved by framing the allocation as a multi-armed bandit problem, where the number of batches assigned to each strategy is adjusted based on observed batch quality. Furthermore, by measuring the similarity of selected candidate batches, we could assess redundancy among ensemble members and encourage more independent selection, increasing the likelihood of retaining diverse, informative instances. Such mechanisms would allow BoSS to automatically focus computational resources on the most effective and independent strategies for a given dataset and cycle, potentially improving both efficiency and performance.

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

## A   Practical Recommendations

Our oracle BoSS is best employed for evaluating and comparing AL strategies. It allows researchers to pose questions such as "*How far away is my newly proposed AL strategy from the optimal performance?*" or "*At which stage of AL (e.g., early vs. late cycles) does my strategy underperform?*". If a new AL strategy is close to our oracle, it is a reliable indicator for a well working selection. Vice versa, a large gap implies that the strategy might struggle and that there may be potential for improvement.

When employing BoSS in experiments, we recommend a simple procedure without needing to implement each AL strategy listed in Table 1. Specifically, if an author has developed a novel AL strategy and intends to evaluate it alongside four additional state-of-the-art strategies, we suggest simply using the already implemented AL strategies for candidate batch generation. Any further hyperparameters can be set to the default values presented in this work. Regarding the budget for a given dataset, we recommend determining it by performing Random sampling until the learning curve reaches convergence. This ensures that both low- and high-budget scenarios are accounted for in the evaluation, which can potentially reveal issues in exploration and exploitation. Furthermore, although our experiments demonstrated minimal sensitivity to the choice of loss function, employing an alternative loss function may be beneficial if the evaluation metric significantly differs from classification accuracy.

To illustrate this, we consider an experimental scenario where we assume a novel AL strategy (i.e., DropQuery) and seek to evaluate its performance against established state-of-the-art strategies (i.e., BADGE, BAIT, TypiClust). Accordingly, we construct BoSS by defining the set $\mathcal{S}$ to include Drop-Query, BADGE, BAIT, TypiClust, and Random sampling, with 10 candidate batches per strategy. Comparing the new AL strategy DropQuery with BoSS in Fig. 6 provides multiple insights: On CIFAR-10, a clear performance difference emerges

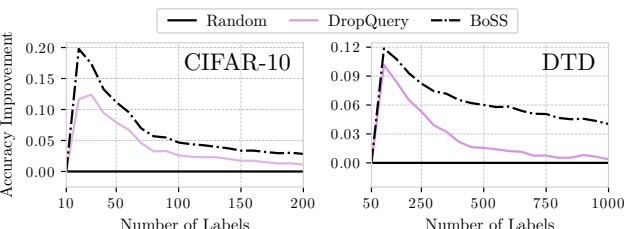

Figure 6: Explanatory plot of the "new" strategy DropQuery in relation to both Random and BoSS.

in the initial cycles, after which both DropQuery and BoSS behave similarly. This suggests DropQuery might struggle to identify influential instances early on but continues to perform well for the rest of the experiment. On DTD, a more complex dataset, although the initial performance gap is smaller, the gap between DropQuery and BoSS continuously increases in subsequent cycles. This indicates that while effective in the beginning, exploitation may not work properly in later cycles.

## B   Varying Candidate Pool Size

Here, we examine the influence of varying the size of the candidate pools from which candidate batches are selected. In general, sampling candidate pools enables deterministic AL selection strategies to generate multiple distinct candidate batches. Moreover, selecting batches from smaller subsets rather than from the entire unlabeled pool improves computational efficiency. However, choosing the appropriate subset size presents a trade-off. On one hand, the subset must be sufficiently large to ensure the presence of influential instances. On the other hand, overly large subsets may re-

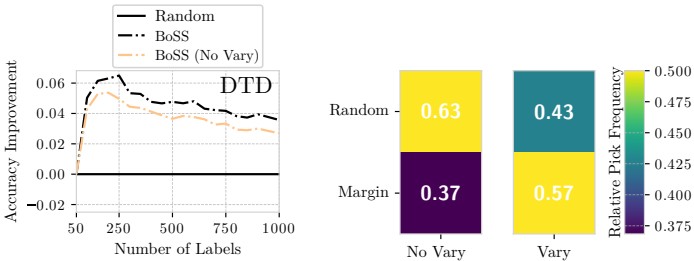

Figure 7: Relative learning curves and pick frequencies with (Vary) and without (No Vary) varying the subset size for candidate batch generation.

duce randomness, resulting in deterministic strategies repeatedly selecting similar candidate batches. Smaller subset sizes introduce more randomness into batch selection, whereas larger subset sizes emphasize the intrinsic characteristics of the employed AL strategies. Thus, identifying the optimal subset size is difficult.

For this reason, we vary the subset size of candidate pools in BoSS. When the subset size is small, candidate batches exhibit more randomness. In contrast, when the subset size is large, the selection of candidate batches is increasingly driven by the heuristics of the employed selection strategies. Due to the performance-based view of BoSS, we ensure that low-performing candidate batches from unsuitable subset sizes do not influence the oracle's overall AL performance. Figure 7 demonstrates this effect. For this experiment, we include only Random and Margin as BoSS's selection strategies and focus on DTD with the DINOv2-ViT-S/14 model. We observe that without varying candidate pool sizes, BoSS remains strongly biased toward randomly sampled batches. In contrast, varying pool sizes shifts selection toward Margin and yields an increase in performance.

## C  Analysis of AL Strategies: Uncertainty vs. Representativeness

In addition to its strong performance, BoSS's performance-based selection of candidate batches enables us to assess which AL strategy is most effective at each cycle. Specifically, by looking at which candidate batch was selected by BoSS, we can identify whether a particular AL strategy excels early, later, or across all stages of the AL process. To illustrate this, we run BoSS on CIFAR-100 and Food101 with the DINOv2-ViT-S/14 model using three selection strategies for candidate batch generation. We include Random sampling, the representativeness-based strategy TypiClust, and the uncertainty-based strategy Margin. Following the intuitions from (Kottke et al., 2021; Hacohen & Weinshall, 2023), early cycles should benefit from representative instances to capture the task's underlying distribution, while later cycles should benefit from uncertain instances. Figure 8 shows the average pick frequency of BoSS over ten runs on both datasets. The selection pattern reveals a clear preference for representative candidate batches in the first three cycles, as TypiClust is primarily picked at that stage. Contrary to the intuitions, however, BoSS does not exclusively focus on uncertain instances later on but continues to select a mix of random, uncertain, and representative batches. This suggests that either Margin may be less effective at identifying truly challenging instances on these datasets or that the intuition that AL should pivot solely to uncertain instances may be overly simplistic. Furthermore, the fact that randomly sampled candidate batches are chosen suggests that none of the selection strategies provide influential batches at a given stage.

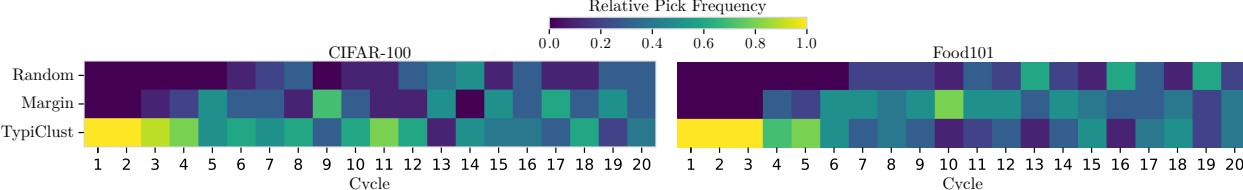

Figure 8: Average pick choices of BoSS with three selection strategies on CIFAR-100 and Food101.

## D  Pick Choices Per Dataset

Supplementing $RQ_4$, we present the average relative pick frequencies for each dataset. Figure 9 shows how often each selection strategy was chosen as the best candidate batch across datasets. Three main insights emerge. First, although DropQuery* and BAIT achieve the highest pick frequencies on several datasets, every selection strategy contributes influential candidate batches at various stages of the AL process. This underscores the value of including each selection strategy in BoSS. Second, not only does every selection strategy get selected at least once, confirming that no single strategy dominates an entire AL cycle, but there is also no consistently preferred selection strategy across all datasets. This highlights that the ensemble of AL strategies itself is critical for maintaining strong, dataset-agnostic performance. Third, for datasets with larger batch sizes ($\geq 100$), pick frequencies mostly concentrate on two selection strategies. This pattern suggests that the other AL strategies struggle to propose effective candidate batches as dataset complexity grows. Moreover, since DropQuery* cannot be simply applied in AL (labels in the unlabeled pool are unavailable), BAIT emerges as a promising alternative in that context.

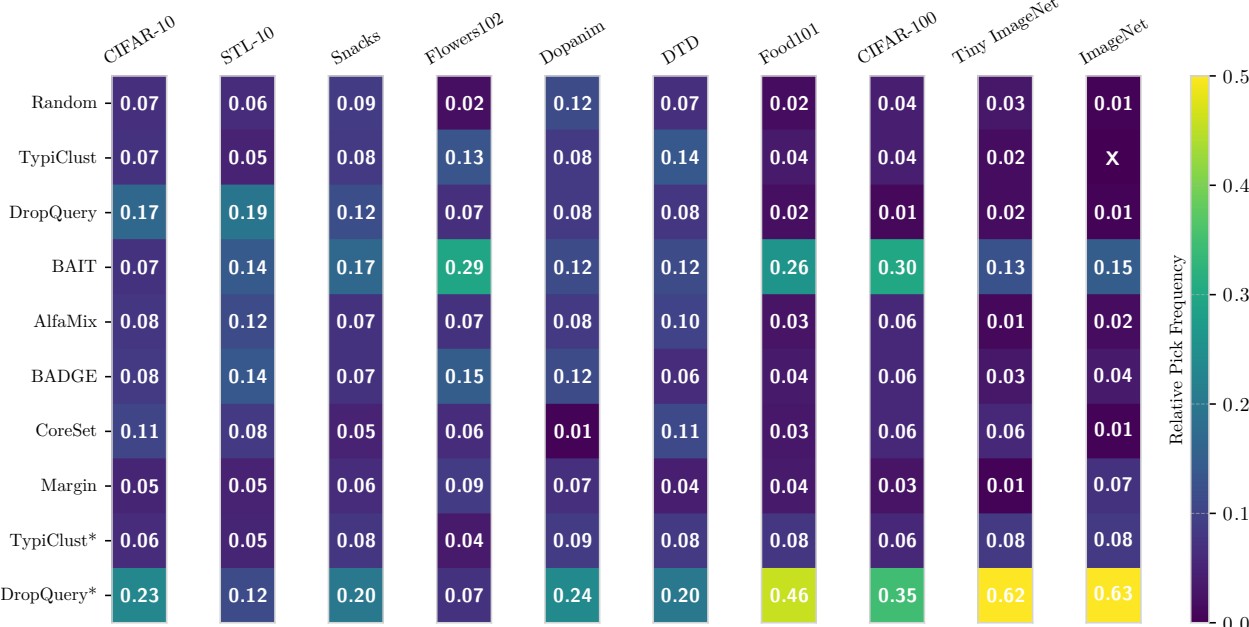

Figure 9: Relative pick frequencies of selection strategies by BoSS per dataset averaged over cycles.

# E   Minimal Oracle

In Section 8, we investigated various factors influencing the performance of BoSS and settled for a good trade-off between runtime and effectiveness. Here, we aim to examine how BoSS's performance decreases when prioritizing runtime only. Therefore, we introduce three different runtime-optimized variants of our original oracle, namely BoSS (S) with $T = 50$ and 25 retraining epochs, BoSS (XS) with $T = 25$ and 10 retraining epochs, and BoSS (XXS) with $T = 10$ and 5 retraining epochs. The results in Fig. 10 show that while these runtime-optimized variants yield slightly reduced performance, BoSS still performs reasonably. Especially considering the simpler dataset CIFAR-10, even BoSS (XXS) still yields the best performance when compared to all considered state-of-the-art AL strategies. Thus, we want to emphasize that the values chosen in Section 8 are guideline values and that BoSS can also work well when runtime needs to be significantly reduced.

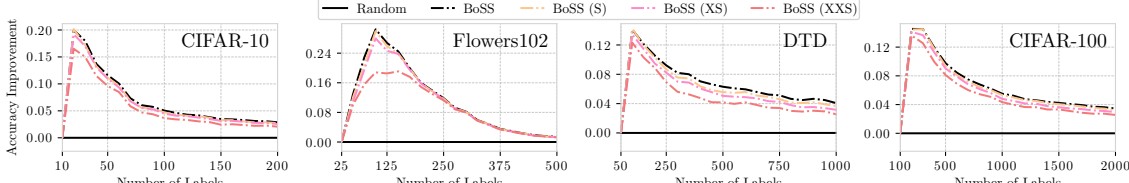

Figure 10: Relative learning curves of BoSS and its runtime-optimized variants.

# F   Supplementary Absolute Learning Curves

In addition to the relative learning curves presented in the main part of the paper, we also show the associated absolute learning curves here. Figure 11 depicts the absolute learning curves that correspond to the relative curves reported in Fig. 2. Similarly, in Fig. 13 we report the corresponding absolute learning curves of the state-of-the-art experiments from Fig. 3. Additionally, we also report the absolute learning curves of all oracle strategies with default hyperparameters in Fig. 12.

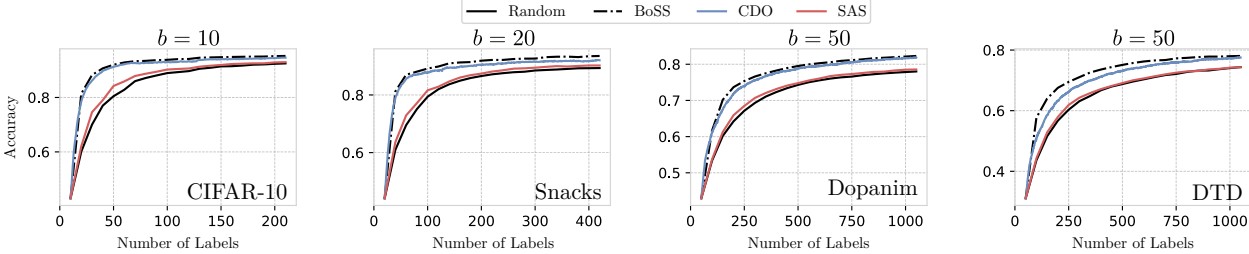

Figure 11: Absolute learning curves of oracle strategies with aligned runtimes using DINOv2-ViT-S/14.

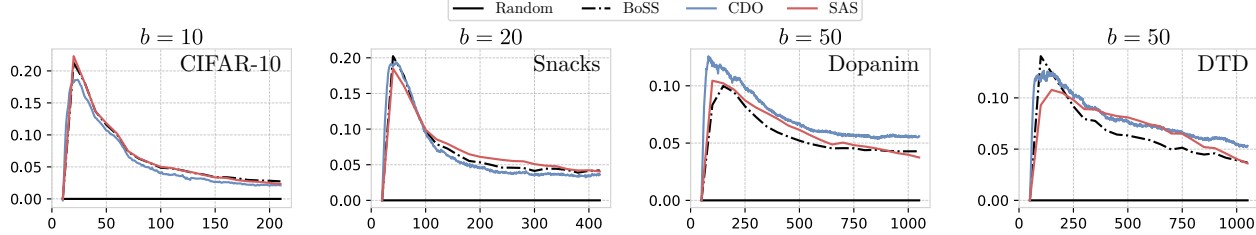

Figure 12: Relative learning curves of oracle strategies with default hyperparameters using DINOv2-ViT-S/14.

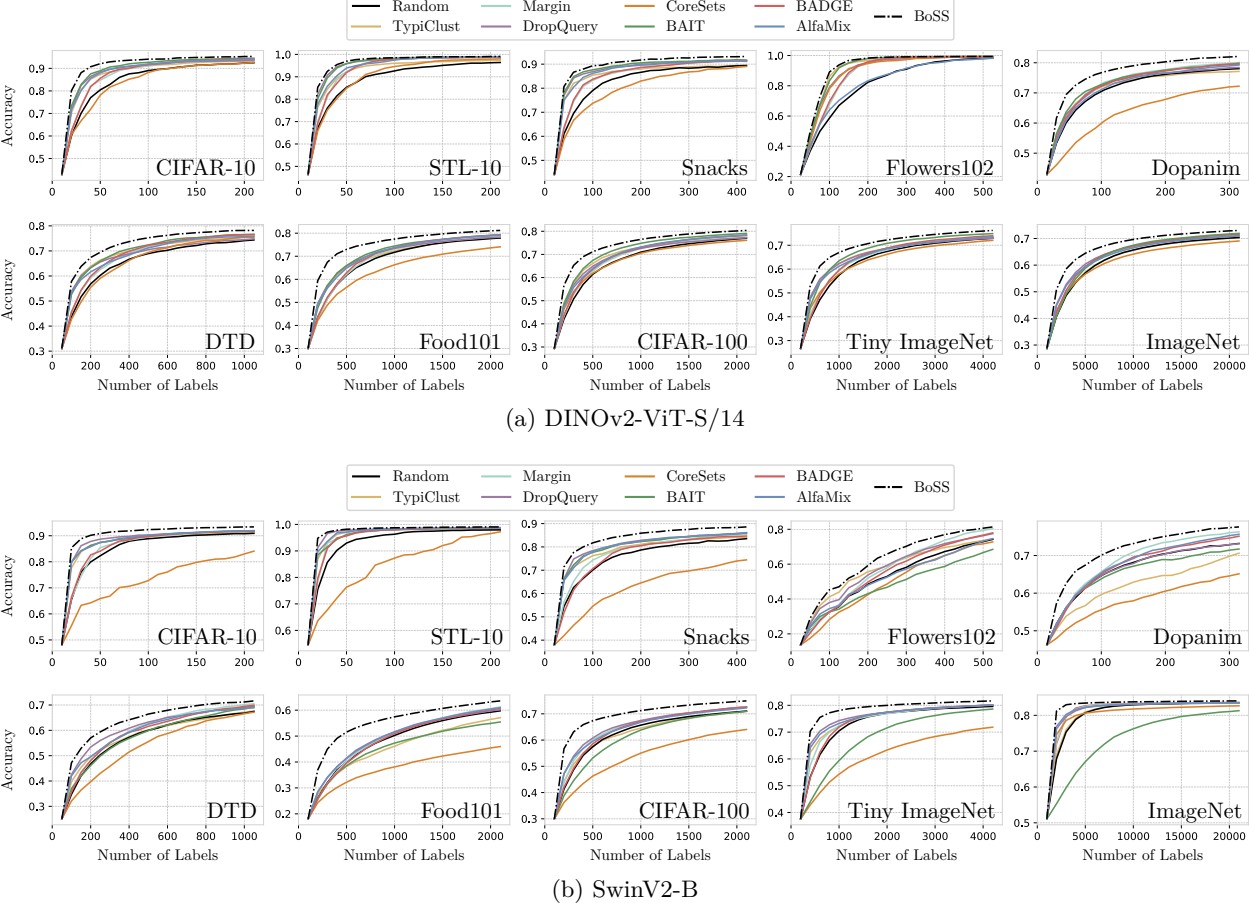

Figure 13: Absolute learning curves achieved by BoSS and state-of-the-art selection strategies at each annotation cycle for different pretrained models.

# G Budget-Regime Analysis with DINOv2

Here, we additionally analyze the area under learning curves (AULC) across all budget regimes for the DINOv2-ViT-S/14 backbone with the goal to facilitate an easy comparison to related work. Note that the reported AULC values correspond to specific segments of the learning curve: low (cycle 1–7), mid (cycle 7–14), and high (cycle 14–20), allowing for a more granular assessment of performance at different annotation budgets.

We observe clear differences between the selection strategies across all budgets. At **low-budget** (Table 11), performance differences are substantial. Simple baselines like Random or CoreSets yield weak results, while more sophisticated strategies such as DropQuery, BAIT, and BoSS clearly outperform them across datasets. At **mid budget** (Table 12), the gap between strategies narrows, but BoSS remains the leading approach, followed by BAIT and BADGE. Margin and TypiClust provide moderate improvements over Random sampling. At **high budget** (Table 13), performances saturate and differences between strategies become smaller. Nevertheless, BoSS consistently yields the highest AULC values across all datasets, with BAIT and BADGE being strong feasible strategies. **Overall**, BoSS dominates in all budget regimes, highlighting its role as an oracle. Among practically usable strategies, BAIT and BADGE are the most competitive, especially at medium and high budgets, while CoreSets tends to underperform throughout.

Table 11: AULC for the low-budget regime across vision datasets using DINOv2 features.

|  | CIFAR-10 | STL-10 | Snacks | Flowers102 | Dopanim | DTD | Food101 | CIFAR-100 | Tiny ImageNet | ImageNet |
|---|---|---|---|---|---|---|---|---|---|---|
| Random | 0.713 | 0.763 | 0.707 | 0.548 | 0.612 | 0.531 | 0.538 | 0.531 | 0.496 | 0.500 |
| Margin | 0.745 | 0.813 | 0.740 | 0.630 | 0.628 | 0.550 | 0.539 | 0.544 | 0.513 | 0.513 |
| CoreSets | 0.693 | 0.759 | 0.668 | 0.680 | 0.525 | 0.519 | 0.503 | 0.543 | 0.508 | 0.501 |
| BADGE | 0.748 | 0.810 | 0.742 | 0.640 | 0.626 | 0.555 | 0.546 | 0.550 | 0.512 | 0.519 |
| TypiClust | 0.777 | 0.825 | 0.776 | 0.721 | 0.629 | 0.587 | 0.571 | 0.577 | 0.546 | 0.503 |
| DropQuery | 0.789 | 0.862 | 0.792 | 0.693 | 0.629 | 0.584 | 0.576 | 0.567 | 0.546 | 0.530 |
| BAIT | 0.796 | 0.857 | 0.799 | 0.717 | 0.638 | 0.586 | 0.575 | 0.584 | 0.545 | 0.511 |
| AlfaMix | 0.783 | 0.836 | 0.780 | 0.580 | 0.617 | 0.573 | 0.569 | 0.561 | 0.537 | 0.523 |
| BoSS | 0.829 | 0.875 | 0.813 | 0.744 | 0.677 | 0.619 | 0.643 | 0.626 | 0.587 | 0.569 |

Table 12: AULC for the mid-budget regime across vision datasets using DINOv2 features.

|  | CIFAR-10 | STL-10 | Snacks | Flowers102 | Dopanim | DTD | Food101 | CIFAR-100 | Tiny ImageNet | ImageNet |
|---|---|---|---|---|---|---|---|---|---|---|
| Random | 0.893 | 0.931 | 0.869 | 0.889 | 0.747 | 0.695 | 0.725 | 0.715 | 0.681 | 0.662 |
| Margin | 0.916 | 0.979 | 0.888 | 0.980 | 0.765 | 0.722 | 0.736 | 0.732 | 0.687 | 0.674 |
| CoreSets | 0.886 | 0.949 | 0.835 | 0.967 | 0.658 | 0.701 | 0.671 | 0.711 | 0.669 | 0.647 |
| BADGE | 0.919 | 0.979 | 0.888 | 0.981 | 0.760 | 0.726 | 0.744 | 0.736 | 0.695 | 0.677 |
| TypiClust | 0.913 | 0.961 | 0.882 | 0.981 | 0.748 | 0.721 | 0.729 | 0.736 | 0.693 | 0.662 |
| DropQuery | 0.919 | 0.981 | 0.900 | 0.973 | 0.756 | 0.709 | 0.740 | 0.731 | 0.692 | 0.675 |
| BAIT | 0.930 | 0.984 | 0.906 | 0.988 | 0.763 | 0.732 | 0.750 | 0.753 | 0.712 | 0.680 |
| AlfaMix | 0.922 | 0.976 | 0.898 | 0.893 | 0.753 | 0.718 | 0.744 | 0.738 | 0.690 | 0.670 |
| BoSS | 0.942 | 0.986 | 0.919 | 0.989 | 0.793 | 0.756 | 0.779 | 0.769 | 0.727 | 0.700 |

Table 13: AULC for the high-budget regime across vision datasets using DINOv2 features.

|  | CIFAR-10 | STL-10 | Snacks | Flowers102 | Dopanim | DTD | Food101 | CIFAR-100 | Tiny ImageNet | ImageNet |
|---|---|---|---|---|---|---|---|---|---|---|
| Random | 0.919 | 0.959 | 0.889 | 0.972 | 0.774 | 0.734 | 0.770 | 0.759 | 0.720 | 0.696 |
| Margin | 0.936 | 0.988 | 0.910 | 0.993 | 0.793 | 0.760 | 0.777 | 0.771 | 0.726 | 0.707 |
| CoreSets | 0.919 | 0.977 | 0.879 | 0.989 | 0.710 | 0.746 | 0.726 | 0.751 | 0.710 | 0.682 |
| BADGE | 0.938 | 0.987 | 0.911 | 0.993 | 0.789 | 0.759 | 0.784 | 0.775 | 0.731 | 0.708 |
| TypiClust | 0.930 | 0.971 | 0.892 | 0.990 | 0.768 | 0.749 | 0.770 | 0.766 | 0.725 | 0.695 |
| DropQuery | 0.933 | 0.986 | 0.911 | 0.990 | 0.778 | 0.740 | 0.774 | 0.763 | 0.722 | 0.704 |
| BAIT | 0.942 | 0.989 | 0.916 | 0.993 | 0.790 | 0.761 | 0.784 | 0.783 | 0.742 | 0.711 |
| AlfaMix | 0.937 | 0.987 | 0.912 | 0.969 | 0.783 | 0.754 | 0.783 | 0.775 | 0.726 | 0.702 |
| BoSS | 0.951 | 0.990 | 0.930 | 0.993 | 0.816 | 0.778 | 0.805 | 0.797 | 0.754 | 0.724 |

# H   Additional Results on Text Classification Tasks

We additionally evaluated BoSS on four text classification datasets oriented on the text AL benchmark (Rauch et al., 2023). In particular, we include the large-scale news classification dataset AG-News (Zhang et al., 2015) ($K = 4$), the medium-cardinality ontology classification dataset DBPedia (Lehmann et al., 2015) ($K = 14$), and the high-cardinality intent detection dataset Banking (Casanueva et al., 2020) ($K = 77$). Furthermore, to extend the evaluation beyond 100 classes, we also include the intent classification dataset Clinc (Larson et al., 2019) ($K = 150$). Similar to the setup in the main paper, we extracted features using a pretrained model. Here, we employ the MiniLM language model (Reimers & Gurevych, 2019), though any model pretrained on text data could have been used to extract features. Figure 14 reports absolute and relative learning curves over 10 repetitions. The results confirm that BoSS also achieves strong performance in the text domain, reinforcing its applicability beyond vision.

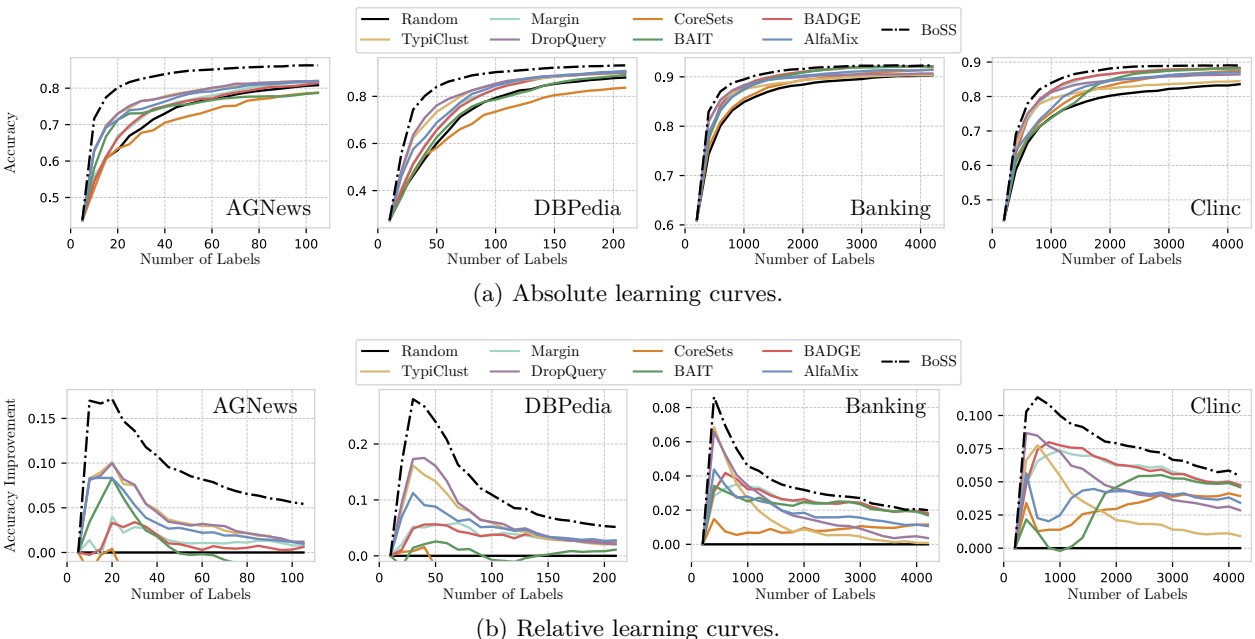

(a) Absolute learning curves.

(b) Relative learning curves.

Figure 14: Learning curves of BoSS and state-of-the-art selection strategies across annotation cycles on four text classification datasets using MiniLM features.

# I   Analyzing the Impact of Supervision in BoSS

To analyze the effect of supervision within BoSS, we conduct two controlled variants that progressively reduce the amount of supervised information available during batch selection.

**BoSS without supervised strategies:** In this setting, we restrict the candidate batches to only be generated by unsupervised AL strategies, i.e., methods that do not rely on label information at selection time. This allows us to examine whether the performance of BoSS persists when the supervised signal is entirely removed from the candidate generation process. **BoSS with approximate supervision:** Here, instead of using ground-truth labels for retraining or performance evaluation, we infer them by an approximate distribution. Specifically, we transform the original formulation in Eq. (3) to be usable with no labels

$$\underset{\mathcal{B} \subset \mathcal{U}}{\arg\min} \, \mathbb{E}_{p(\boldsymbol{x},y)}\big[\ell\big(y, p(y|\boldsymbol{x}, \mathcal{L}^+)\big)\big] \approx \underset{\mathcal{B} \subset \mathcal{U}}{\arg\min} \, \mathbb{E}_{p(\boldsymbol{y}_{\mathcal{B}}|\boldsymbol{X}_{\mathcal{B}})}\mathbb{E}_{p(\boldsymbol{x})}\mathbb{E}_{p(y|\boldsymbol{x})}\big[\ell\big(y, p(y|\boldsymbol{x}, \mathcal{L}^+)\big)\big] \tag{6}$$

$$\approx \underset{\mathcal{B} \subset \mathcal{U}}{\arg\min} \, \mathbb{E}_{p(\boldsymbol{y}_{\mathcal{B}}|\boldsymbol{X}_{\mathcal{B}}, \boldsymbol{\omega}^\star)}\mathbb{E}_{p(\boldsymbol{x})}\mathbb{E}_{p(y|\boldsymbol{x}, \boldsymbol{\omega}^\star)}\big[\ell\big(y, p(y|\boldsymbol{x}, \mathcal{L}^+)\big)\big], \tag{7}$$

where we approximate $p(\boldsymbol{y}_\mathcal{B}|\boldsymbol{X}_\mathcal{B})$ and $p(y|\boldsymbol{x})$ with a model $\boldsymbol{\omega}^\star$ trained on the entire training dataset. This approximation represents a more realistic oracle that relies on predictive uncertainty rather than perfect supervision. We employ the DINOv2-ViT-S/14 backbone with the same experimental setup as in the main paper.

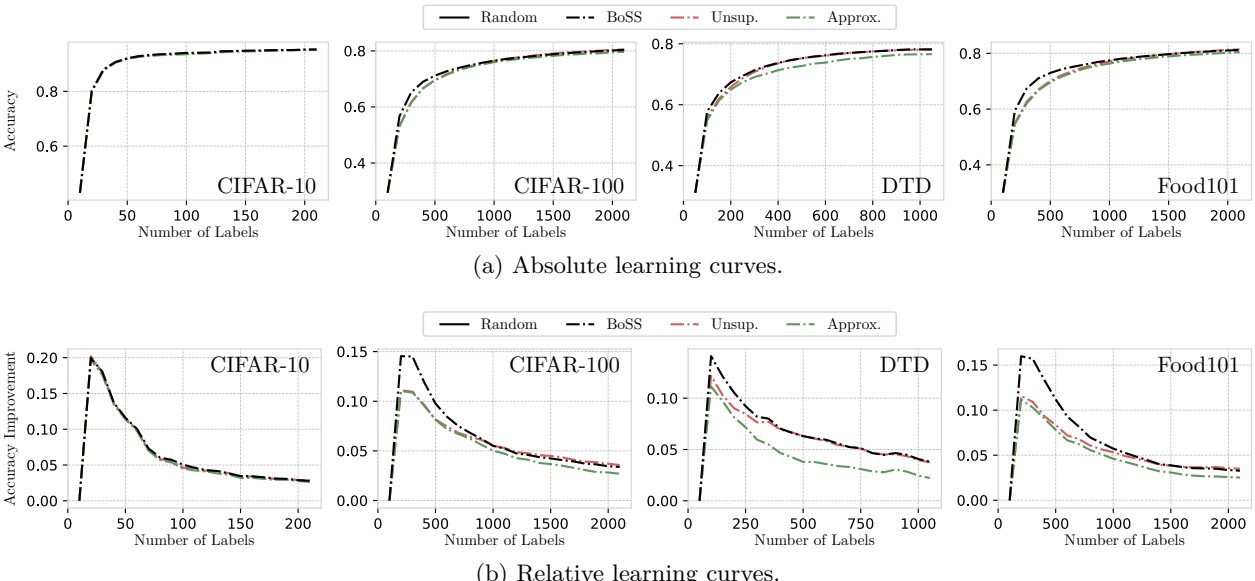

(a) Absolute learning curves.

(b) Relative learning curves.

Figure 15: Learning curves of BoSS and its variants with reduced supervision: without supervised strategies (Unsup.) and with approximate supervision (Approx.).

Based on the learning curves in Fig. 15, we see that the performance gap between different supervision levels of BoSS varies across datasets. On CIFAR-10, both BoSS without supervised strategies (Unsup.) and the approximate variant (Approx.) achieve accuracies comparable to the fully supervised oracle, indicating that label information provides minimal additional benefit. However, on more challenging datasets with many classes (CIFAR-100 and Food101), supervised strategies seem to play a more important role, particularly during the early stages of AL. However, by the end of the AL process, the unsupervised variant is on par with the fully supervised oracle, suggesting that the value of supervision diminishes as the labeled pool grows. Notably, on DTD, a dataset where DINOv2-ViT-S/14 appears less well-suited due to substantial domain mismatch, the approximate supervision variant (Approx.) exhibits degraded performance. We believe this is due to areas with high aleatoric uncertainty in the embedding space that yield incorrect labels. To potentially mitigate this, we suggest employing predictive models that explicitly disentangle uncertainties. Overall, our experiment shows that supervision can play an important role in BoSS when datasets are noisy or exhibit significant domain shift. For future work, we believe that learning a candidate batch selection policy represents a promising research direction. This way, it might be possible to imitate BoSS's behavior by directly predicting the most promising candidate batch without requiring explicit label access.

