# OpenReview forum: "BoSS: A Best-of-Strategies Selector as an Oracle for Deep Active Learning"
_TMLR — Accepted by TMLR_

### Review · Reviewer_h6PP · 2025-09-19

**Summary Of Contributions:**

The paper proposes BoSS, an oracle-style upper baseline for deep active learning (AL). In each AL cycle, BoSS (i) constructs candidate batches using an ensemble of existing selection strategies, (ii) estimates performance for each candidate by retraining only the final linear layer on frozen features, and (iii) chooses the batch that yields the largest test-set accuracy gain. The method is positioned as a scalable oracle for large models/datasets where prior oracles, SAS (simulated annealing) and the Cross-Domain Oracle (CDO), struggle to scale. Experiments on 10 image datasets and two strong backbones (DINOv2-ViT-S/14 and SwinV2-B) show BoSS outperforms CDO/SAS under matched runtime and consistently upper-bounds SOTA AL strategies across cycles; the gaps are largest on large-class datasets (e.g., CIFAR-100, Tiny ImageNet, ImageNet) .

**Audience:**

Yes

**Audience Explanation:**

The method provided in this paper is accurate and much scalable than existing methods, which has the potential to be widely used.

**Claims And Evidence:**

Yes

**Claims Explanation:**

**Strength**
1. The paper is in general well-written, with motivation, related works, and methodology clearly explained and illustrated.
2. The author provided substantial results on various vision datasets spanning up to 1000 classes and batches up to 1000, with two strong backbones (DINOv2-ViT-S/14; SwinV2-B).
3. The scalability of BOSS is well-justified and the performance is promising. It shows clear better results than SAS/CDO under small runtime budget and remains comparable with them in the default setting.
4. Ablations on $T$, loss, and proxy epochs (5 to 200) support the proxy choice and show diminishing returns beyond ~10 batches/strategy and ~50 proxy epochs

**Weakness**
1. The paper does not consider domains other than vision, while related works, e.g. Werner et al. 2024 also considered text/tabular domains.
2. More explicit budget-regime analysis would be helpful, e.g. by defining and slicing the AULC by early, mid, and late regime as Hacohen et al. 2022.
3. Some experimental details, e.g., how is $k_{max}$ chosen, how sensitive is the method to it, hardware resources, are not clear to me.

**Requested Changes:**

1. Provide one additional domain for BOSS.
2. More experimental details as mentioned above.

---

> ### Author Response · Authors · 2026-02-18
>
> We thank the reviewer for the constructive feedback and are glad that the strengths of our paper were recognized. At the same time, the comments helped us clarify experimental details and refine our evaluation. Below, we address each of the mentioned weaknesses individually.
>
> > The paper does not consider domains other than vision, while related works, e.g. Werner et al. 2024 also considered text/tabular domains.
> >
>
> We agree that restricting the evaluation to vision tasks is a limitation compared to related works such as Werner et al. (2024). To address this, we broadened our evaluation and included experiments on text classification. Specifically, we employed the MiniLM language model [1] to extract features and followed the same experimental setup as in the main manuscript. We ran these experiments on four text datasets that were also included in the AL benchmark of [2]. The results show that BoSS also achieves strong performance in the text domain, reinforcing the findings from the main paper. The detailed experiments are provided in Appendix H.
>
> [1] Reimers, N. & Gurevych, I. (2019). Sentence-BERT: Sentence embeddings using Siamese BERT-networks. In *EMNLP*.
>
> [2] Rauch, L., et al. (2023). Activeglae: A benchmark for deep active learning with transformers. In *ECML PKDD*.
>
> > More explicit budget-regime analysis would be helpful, e.g. by defining and slicing the AULC by early, mid, and late regime as Hacohen et al. 2022.
> >
>
> We agree with the reviewer that the paper benefits from a more explicit budget-regime analysis. Following this suggestion, we now provide AULC values sliced by early, mid, and late budget regimes in Appendix G, along with a discussion. We hope this addition facilitates easier comparison for future work.
>
> > Some experimental details, e.g., how is $k_{\text{max}}$ chosen, how sensitive is the method to it, hardware resources, are not clear to me.
> >
>
> We thank the reviewer for pointing this out and agree that some experimental details require clarification. In general, the choice of $k_{\text{max}}$ is meant to define a representative subset of the dataset. For datasets with relatively few classes, such as CIFAR-10, we fixed $k_{\text{max}} = 1000$, which i) preserves most of the dataset information while ii) still allows fast selection by the proposed strategies. For more complex datasets such as ImageNet, we increased its value to $k_{\text{max}}= 2500$. In our experiments, performance was not sensitive to this parameter, which is why we did not conduct a separate ablation study. We now include a discussion on how to set $k_{\text{max}}$ in Section 5.1. Furthermore, as suggested, we clarified the hardware resources used for our experiments in the experimental setup.
>
> We hope that our clarifications and additional experiments address all of the reviewer’s concerns. We thank the reviewer once again for the valuable feedback, which helped us improve the quality and clarity of the paper.

---

### Review · Reviewer_d3VS · 2025-10-01

**Summary Of Contributions:**

This manuscript proposes a best-of-strategies selector (BOSS) to attempt to evaluate how well current active learning strategies are doing compared to an upper bound based on using supervised information.  This is an interesting idea that could be used to evaluate when and how each strategy is best used, potentially helping us choose from an ensemble in active learning strategies.  There is a claim that this upper bound identifies that "ii) state-of-the-art AL strategies have significant room for improvement, especially in large-scale datasets with many classes."  However, this claim is not fully supported by the analysis, as it is not clear that this would be achievable by any method, as the upper bound is constructed with supervised labels, which would never be available in active learning.

**Audience:**

Yes

**Audience Explanation:**

Active learning is an active area of research, and knowing how good the models could possibly be is a useful set of knowledge to help explore in which situations the gaps are largest.  These results, if the points above are addressed, could be helpful to that field to guide researchers to the areas with the greatest potential improvements.  I'd appreciate if the authors explored this point further as well.

**Broader Impact Concerns:**

None.

**Claims And Evidence:**

No

**Claims Explanation:**

This manuscript blurs the distinction between “oracle strategies” and “achievable upper bound” that's useful as a diagnostic tool, which to me is the biggest issue.  A main claimed contribution is that "ii) state-of-the-art AL strategies have significant room for improvement, especially in large-scale datasets with many classes"; however, it is only clear that BOSS is an upper bound, not that there is significant room for improvement, as it is not clear how much of this gap would be closable without the supervised information used in BOSS.  In order to address this, the authors should be clear that BoSS is an infeasible oracle that is not necessarily achievable.  In particular, the authors should avoid implying that the gap equals attainable headroom, as I do not see a clear scientific justification for that claim.  In order to address this, the authors need to quantify how much of the gap is due to true oracle knowledge vs. fixable heuristic weakness.  In other words, how much does knowing the labels during selection impact the performance?

It would be extremely helpful to try to estimate (1) what is achievable gap versus (2) what part of the gap is due to the supervised information being used in the construction of the upper bound.  This is a challenging problem that does not have a clear solution to me.  As is, I do not think that one of the main claims is fully supported by the analysis and experiments.

As a second point, I am surprised by the inclusion of the pretrained SwinV2-B model, as it is pre-trained on supervised Imagenet labels.  While the final layer is rewritten, it seems problematic that imagenet is being evaluated on a model that was pre-trained on imagenet. The authors need to discuss this at greater length for potential issues of data leakage, and how that interplays with the assumptions made in the model training.  Specifically, this seems like the best case scenario for the strategy claimed as, "Specifically, we freeze the feature
extractor’s parameters ϕ and only retrain the final linear layer θ."  If the authors want to include these results, they need to explain how they are appropriate for inclusion given the potential data leakage problem?

**Requested Changes:**

A fundamental issue is the potential utility of the upper bound and whether it is achievable. Again, the authors need to estimate how much of the gap is due to true oracle knowledge vs. fixable heuristic weakness, which is necessary for evaluating the contribution of this work and the true remaining scientific gap.

A second issue is that a greater discussion on the utility/appropriateness of SwinV2-B and its conclusions should be added.

---

> ### Author Response · Authors · 2026-02-18
>
> We would like to thank the reviewer for the thoughtful and detailed comments, as well as for the time and effort invested. The feedback has been very helpful in clarifying our assumptions and improving the presentation, and we have revised the paper accordingly.
>
> We carefully address each point below.
>
> > This manuscript blurs the distinction between “oracle strategies” and “achievable upper bound” that's useful as a diagnostic tool, which to me is the biggest issue
> >
>
> This is an important observation. Our original assumption was that if there exists a batch yielding the highest performance (regardless of how it can be evaluated), then, in principle, a selection strategy should be able to provide it. Based on this reasoning, we initially presented BoSS as an upper baseline for AL strategies. We now recognize, however, that this assumption does not hold in practice and certainly not without access to ground-truth information. **In line with the reviewer’s concern, we have revised our claims accordingly. BoSS is not considered an upper baseline anymore, and we clarify that it represents an oracle rather than an attainable benchmark.** Additionally, to emphasize this further, **we also included a limitations section which discusses exactly this problem.**
>
> > In particular, the authors should avoid implying that the gap equals attainable headroom, as I do not see a clear scientific justification for that claim.
> >
>
> At the same time, we would like to highlight an important property of BoSS: it can be designed as an oracle that assesses batches that were proposed by unsupervised AL strategies. Consequently, when BoSS selects a batch that yields the highest performance gain, this indicates that the candidate strategy was indeed capable of generating such a batch. While we carefully revised our claims, we emphasize that the observed performance gap should neither be interpreted as fully attainable nor attributed solely to the use of supervised information. Rather, we argue that the gap reflects a combination of (i) genuine weaknesses in current AL heuristics that could, in principle, be addressed, and (ii) performance advantages arising only from access to ground-truth labels during selection. **We have also made this point explicit in the limitations section and added experiments in the Appendix, analyzing BoSS when applied solely in combination with unsupervised strategies.**
>
> > This is a challenging problem that does not have a clear solution to me.  As is, I do not think that one of the main claims is fully supported by the analysis and experiments.
> >
>
> We agree that this is a challenging problem and that a definitive solution is difficult to obtain. To address this concern, we revised our claims to ensure they are fully supported by the analysis. In addition, to partially assess the impact of reducing the supervised signal, we conducted further experiments examining how BoSS performs under a reduced-supervision setting. **These results, referenced in the Limitations section, are provided in the Appendix.**
>
> We believe that by weakening our claims, adding a dedicated limitations section, and including additional experiments in the Appendix, we have adequately addressed this concern. Thank you again for bringing up this really essential point. We are confident that our claims are now accurately supported by the experiments and analysis presented in the paper.

---

> ### Author Response · Authors · 2026-02-18
>
> > As a second point, I am surprised by the inclusion of the pretrained SwinV2-B model, as it is pre-trained on supervised Imagenet labels. While the final layer is rewritten, it seems problematic that imagenet is being evaluated on a model that was pre-trained on imagenet. The authors need to discuss this at greater length for potential issues of data leakage, and how that interplays with the assumptions made in the model training. Specifically, this seems like the best case scenario for the strategy claimed as, "Specifically, we freeze the feature extractor’s parameters ϕ and only retrain the final linear layer θ."  If the authors want to include these results, they need to explain how they are appropriate for inclusion given the potential data leakage problem?
> >
>
> We appreciate the reviewer’s concern regarding the inclusion of the SwinV2-B. Our motivation for including this backbone was twofold. First, we wanted to demonstrate that BoSS is not restricted to self-supervised pretraining paradigms. Second, SwinV2-B is widely adopted in practice and provides a useful point of comparison. **The ImageNet results with SwinV2-B were reported for completeness.**
>
> We agree that evaluating ImageNet with a backbone pretrained on ImageNet represents a potential best-case scenario. **We explicitly highlight the issue in the Experiment Setup** **and clarify that these results are not fully representative**. Importantly, we want to note that the SwinV2-B experiments complement the self-supervised backbone.
>
> Finally, to further demonstrate the applicability of BoSS, **we include additional results on text data in the Appendix, where a language model is used as the feature extractor.** These evaluations further highlight the robustness and strong performance of BoSS.
>
> We hope that these clarifications, revisions, and additional experiments adequately address the reviewer’s concerns and help resolve potential ambiguities. Thank you again for raising these important points, which we believe have strengthened both the validity and the presentation of our work.

---

> > ### Comment · Reviewer_d3VS · 2026-03-12
> >
> > I appreciate the revision, especially adding the limitations section that makes the claims much clearer.  Overall, this now seems like the claims are now appropriate.
> >
> > I do think that the paper could be strengthened by further efforts to understand how much of the gap between current practice and the oracle strategy is obtainable, but that can be left to future work.

---

### Review · Reviewer_6UCC · 2026-02-05

**Summary Of Contributions:**

This paper introduces an active learning algorithm that functions as an oracle benchmark by integrating existing active learning (AL) methods. This approach can potentially serve as a reference tool for other related research. Specifically, the paper presents an active learning procedure called the Best-of-Strategies Selector (BoSS). The algorithm divides a given query budget, ( B ), into smaller micro-batches, each of size ( b ), and distributes these micro-batches evenly across a pool of selection algorithms. Selection logic is then applied to the strategy-generated batches by querying their ground truth labels and measuring the performance gains they provide. Finally, the filtered examples are merged into a candidate set for retraining.

Empirical studies are conducted on multiple open-source datasets to demonstrate the proposed procedure's merits compared to existing oracle algorithms. Ablation studies are performed to analyze the contribution of each component of the algorithm.

**Audience:**

Yes

**Audience Explanation:**

Active learning remains a important field within machine learning, likely to captivate the interest of the TMLR audience.

**Broader Impact Concerns:**

Not relevant

**Claims And Evidence:**

No

**Claims Explanation:**

There are multiple strength of the proposed method:

1. Empirical studies demonstrate that the procedure can serve as an upper bound for other active learning algorithms within certain labeling budget constraints.
2. The algorithm is efficient in terms of runtime, delivering higher level of quality than the other two oracle methods but running faster.

However, there are several concerns that prevent me from recommending acceptance unless the authors can provide further clarification:

1. I believe that using runtime as a constraint does not create a fair experimental setup. Many external factors can influence runtime results without a highly controlled experimental environment. Additionally, different configurations of baseline methods can result in similar runtimes but different outcomes, yet the authors present only one configuration and its corresponding result. Instead of constraining runtime, the authors should limit the number of data points retrained and evaluated by the each algorithm to constrain the exploration or query budget of each algorithm, allowing for a more apples-to-apples comparison.

2. Some important ablation studies are missing. For example, Table 8 alone does not fully justify the necessity of using ensemble methods. The improvement of might be attributed to the selection-retrain iterations. The authors could consider using CDO as the sole selection strategy in the pool and running a sole-strategy version of BoSS while constraining the query budget. This would help isolate the contribution of the micro-batching operation and provide a clearer assessment of the proposed algorithm's contribution.

3. More explanation is needed regarding the criteria for selecting strategies to be included in the strategy pool. Intuitively, there should be a standard for a selection strategy to be included; otherwise, suboptimal selectors may dominate the computation and limit the samples available to more effective strategies. This leads to a related concern: the micro-batch dispatch procedure seems somewhat coarse. There should be a self-adapt component in the dispatch procedure, allowing effective strategies to gradually receive more micro-batches rather than an equal share with other strategies.

**Requested Changes:**

As previously mentioned, I recommend the author consider the following suggestions:

1. Implement a control mechanism for the query and train-evaluation budget to facilitate a more rigorous comparison.

2. Include an ablation study to substantiate the necessity of using an ensemble approach.

3. Provide a more detailed explanation regarding the criteria for including a strategy in the strategy pools and the rationale behind the even distribution of micro-batches.

---

> ### Author Response · Authors · 2026-02-18
>
> We thank the reviewer for the constructive feedback and the time invested in evaluating our work. The feedback has helped us to considerably improve the presentation of our work.
>
> > I believe that using runtime as a constraint does not create a fair experimental setup. Many external factors can influence runtime results without a highly controlled experimental environment.
> >
>
> We appreciate this important concern. We would like to clarify that all experiments were conducted in a **strictly controlled environment** to minimize external influences on runtime measurements (e.g., by ensuring consistent CPU load, hardware configuration, and Python environment). We have **added details at the end of Section 7.1** summarizing the configuration used.
>
> Furthermore, given the simplicity of CDO and SAS, whose core components consist of model retraining and random instance selection, runtime variability for these methods is inherently low. BoSS runtime is influenced by the strategies included in the ensemble. However, since oracle strategies' runtime is dominated by model retraining, BoSS achieves better efficiency by guiding the search toward promising candidates rather than exhaustively evaluating all candidates.
>
> > Additionally, different configurations of baseline methods can result in similar runtimes but different outcomes, yet the authors present only one configuration and its corresponding result.
> >
>
> We acknowledge that different ensemble configurations can yield varying runtimes and outcomes. In our experiments, we intentionally included all available strategies from Table 1, as our primary aim is to guide the selection through heuristic strategies. Including more strategies enhances robustness across datasets and models. While BoSS could achieve faster runtimes by tailoring the ensemble composition to a specific dataset, we prioritize robustness across settings over dataset-specific optimization. Importantly, our experiments demonstrate that BoSS already scales better than other oracle strategies, even under this non-optimized configuration, where nearly all strategies are included. We also include these details in Section 7.2.
>
> > Instead of constraining runtime, the authors should limit the number of data points retrained and evaluated by the each algorithm to constrain the exploration or query budget of each algorithm, allowing for a more apples-to-apples comparison.
> >
>
> We want to emphasize that we believe the **current experimental setting accurately reflects the efficiency of oracle strategies**. However, we agree that a more detailed analysis of computational cost is beneficial. **Following this suggestion, we have extended our analysis in Section 6** to include the number of training instances processed by each oracle strategy, complementing the number of retraining and evaluation runs.
>
> Based on the batch size and hyperparameters, the total number of training instances processed can be derived as follows:
>
> - **CDO:** $20 (b|\mathcal{L}| + \frac{b(b + 1)}{2})$
> - **SAS:** $1500 A  (|\mathcal{L}| + b)$
> - **BoSS:** $10 |S| (|L| + b)$
>
> Notably, the number of training instances for CDO increases quadratically with the batch size, which is a major bottleneck. In contrast, SAS and BoSS scale linearly. Furthermore, the guided search employed by BoSS enables substantial savings compared to exhaustive retraining approaches. To illustrate this on CIFAR-10, in our experimental setting with $b = 10$ and $|\mathcal{L}| = 50$, CDO processes approximately 11k instances compared to 6k for BoSS. At $b = 50$, this gap widens substantially: CDO requires approximately 75k instances, while BoSS needs only 10k.
>
> It is worth noting that the computational workload of BoSS (specifically the selection of candidate batches) is parallelizable, as each selection operates independently, enabling runtime reduction with appropriate hardware resources. Since the runtime of other strategies is dominated by model retraining, the guided search employed by BoSS offers substantial efficiency gains by avoiding exhaustive evaluation of all candidates.

---

> ### Author Response · Authors · 2026-02-18
>
> > Some important ablation studies are missing. For example, Table 8 alone does not fully justify the necessity of using ensemble methods. The improvement of might be attributed to the selection-retrain iterations. The authors could consider using CDO as the sole selection strategy in the pool and running a sole-strategy version of BoSS while constraining the query budget. This would help isolate the contribution of the micro-batching operation and provide a clearer assessment of the proposed algorithm's contribution.
> >
>
> We appreciate this suggestion and would like to clarify the design rationale behind BoSS. The core idea is to leverage the complementary heuristics captured by different AL strategies, which is an important part of Algorithm 1. We will clarify this reasoning more explicitly in Section 5.1.
>
> Using CDO as the sole selection strategy would fundamentally contradict this objective, as it would reduce BoSS to merely applying CDO repeatedly, which already performs random sampling and batch composition internally. Consequently, it would merely improve selection by considering more instances within CDO itself.
> To address the importance of combining diverse strategies, Table 8 summarizes multiple AL experiments comprising 10 strategies × 10 repetitions = 100 runs. **Each successively added strategy represents a new AL process and contributes to improved performance, despite reducing the number of batches of a potentially well-working strategy.** For instance, when TypiClust is the sole strategy, it generates all 100 batches. Adding Margin halves this to 50, and each subsequent strategy further reduces the number of candidate batches selected per strategy. Despite this, performance consistently improves, demonstrating that strategy diversity outweighs the benefits of concentrating on any single method.
>
> Furthermore, Appendix C presents an ablation study using only TypiClust and Margin as the ensemble. This experiment demonstrates that even with just two strategies, BoSS benefits from their complementary perspectives. TypiClust emphasizes representativeness while Margin focuses on uncertainty. The observed performance gains validate that the ensemble's strength lies in combining diverse selection heuristics rather than in the micro-batching operation alone.
>
> Finally, we want to highlight that a complete evaluation of all possible strategy combinations is **infeasible due to combinatorial constraints**. However, we believe the provided ablations sufficiently demonstrate the value of strategy diversity.
>
> > More explanation is needed regarding the criteria for selecting strategies to be included in the strategy pool. Intuitively, there should be a standard for a selection strategy to be included; otherwise, suboptimal selectors may dominate the computation and limit the samples available to more effective strategies.
> >
>
> We appreciate this concern about strategy composition. As mentioned earlier, the design of BoSS prioritizes including as many diverse perspectives as possible rather than restricting the ensemble to only individually effective strategies. This is motivated by two observations:
>
> 1. As demonstrated in Table 8, successively adding strategies consistently improves or maintains performance. We observe no degradation from including additional strategies, despite reducing the number of batches of a potentially well-working strategy. This suggests that BoSS is robust to weak strategies in the ensemble, as the guided search focuses on promising candidates rather than being dominated by suboptimal selectors.
> 2. A strategy that performs poorly early may still contribute valuable perspectives later. Restricting the ensemble based on individual performance would risk excluding such complementary contributions.
>
> We acknowledge that different strategy configurations may increase computational cost. However, this affects runtime rather than performance. Moreover, BoSS still demonstrates better scalability than other oracles, as shown in our runtime experiments.

---

> > ### Comment · Reviewer_6UCC · 2026-03-03
> > **Response to authors' rebuttal**
> >
> > I appreciate the authors' efforts in addressing my concerns. Firstly, thank you for the additional analysis on the complexity and the further clarification on the running time. This partially helps resolve my concerns. However, I still believe that running time alone may not fully justify the efficiency. My follow-up questions are as follows: Given that CDO and BoSS have similar time complexity and training instance complexity, does CDO benefit from the retraining technique applied only to the final layer? Additionally, if CDO is configured to have a similar running time as BoSS, does its performance significantly decline?
> >
> > Furthermore, I am still concerned about the necessity of adopting the ensemble strategy. The results presented in Table 8 appear to differ from those in Tables 11 to 13. In Table 8, using DINOv2-ViT-S/14 as the backbone and Dopanim as the dataset, BoSS achieves an AULC of $76.52 \pm 0.18$ after including all selection strategies. However, none of the tables from 11 to 13 show similar results with same backbone and dataset, where the result doesn't fall into the error bar given in Table 8. The authors should justify the discrepancy in performance metrics. I believe that maintaining a certain level of consistency in the experimental settings across the ablation study would assist reviewers in making more informed decisions.

---

> > > ### Author Response · Authors · 2026-03-04
> > >
> > > Thank you for the follow-up and active engagement. We appreciate the concerns raised.
> > >
> > > > Given that CDO and BoSS have similar time complexity and training instance complexity, does CDO benefit from the retraining technique applied only to the final layer?
> > > >
> > >
> > > Yes, in all our experiments, CDO, BoSS, and SAS use the same last-layer retraining technique. The only difference between these methods is how they search the candidate space for the best batch. Specifically, CDO randomly samples instances and evaluates them iteratively (non-batch), whereas BoSS leverages selection strategies to guide its search over entire batches, enabling a more efficient and effective exploration of the combinatorial space.
> > >
> > > > Additionally, if CDO is configured to have a similar running time as BoSS, does its performance significantly decline?
> > > >
> > >
> > > Yes. As shown in Figure 2, when AL batch sizes increase such that BoSS and CDO operate under comparable runtimes, CDO's performance notably declines on Dopanim and DTD. This confirms that under equivalent computational budgets, BoSS's batch-aware search provides a meaningful advantage over CDO's single-instance-level sampling.
> > >
> > > > The results presented in Table 8 appear to differ from those in Tables 11 to 13.
> > > …
> > > However, none of the tables from 11 to 13 show similar results with same backbone and dataset, where the result doesn't fall into the error bar given in Table 8. The authors should justify the discrepancy in performance metrics.
> > > >
> > >
> > > This is true and by design. Tables 11–13 provide a regime-specific analysis in which AULC values are computed over only a segment of the learning curve for each regime (low-budget, mid-budget, or high-budget), as requested by Reviewer h6PP. In contrast, Table 8 reports AULC values computed over the entire cycle. Since the two sets of tables evaluate different segments of the learning curve, the resulting values are expected to differ.
> > >
> > > > I believe that maintaining a certain level of consistency in the experimental settings across the ablation study would assist reviewers in making more informed decisions.
> > > >
> > >
> > > We appreciate the concern for consistency. We can assure that all experiments share the same underlying setup. In the camera-ready version, we will also include code that allows reproducing every experiment reported in the paper.
> > >
> > > We have added clarifying additions to our revision and believe these address your remaining concerns. We welcome any further questions or suggestions.

---

> > > > ### Comment · Reviewer_6UCC · 2026-03-04
> > > > **Response to authors' rebuttal**
> > > >
> > > > Thanks for authors' further clarification.  Now I have no further questions.

---

> ### Author Response · Authors · 2026-02-18
>
> > This leads to a related concern: the micro-batch dispatch procedure seems somewhat coarse. There should be a self-adapt component in the dispatch procedure, allowing effective strategies to gradually receive more micro-batches rather than an equal share with other strategies.
> >
>
> We thank the reviewer for this thoughtful suggestion. Indeed, incorporating a self-adaptive component that emphasizes stronger strategies producing high-quality batches is a promising direction. However, the focus of the current paper is to establish the fundamental idea of combining diverse acquisition strategies into an oracle. Introducing an adaptive mechanism would add considerable complexity and warrant thorough investigation in its own right.
>
> Given the importance of this point, we have added a future work paragraph in the conclusion that discusses this mechanism. Specifically, we suggest framing this as a multi-armed bandit problem, dynamically adjusting the number of batches allocated to each strategy based on observed batch quality.
>
> We hope these revisions address all the concerns raised. We thank the reviewer again for their constructive feedback, which has strengthened our work considerably.

---

### Decision · Action_Editor_9yiU · 2026-04-04

**Recommendation:** Accept as is

**Audience:**

Yes

**Audience Explanation:**

Deep active learning is a large and important community within TMLR's audience. The paper is timely and important to guide the research in this direction.

**Claims And Evidence:**

Yes

**Claims Explanation:**

This paper introduces BoSS (Best-of-Strategies Selector), a scalable oracle designed to establish a performance upper bound for deep active learning. By evaluating an ensemble of existing selection strategies and using ground-truth labels to pick the best-performing batch at each training step, BoSS identifies significant gaps where current state-of-the-art methods fall short, particularly in large-scale datasets. Its main contribution is providing a computationally efficient diagnostic tool that helps researchers quantify the "attainable headroom" in active learning and understand which strategies excel at different stages of model training.

The paper was reviewed by three expert reviewers. While initial reviews were critical, the final consensus suggests that the authors successfully addressed major concerns during the revision process. All reviewers uniformly recommends acceptance.